# COVID-19 infection prevention practices among a sample of food handlers of food and drink establishments in Ethiopia

**Atsedemariam Andualem**[1], **Belachew Tegegne**[1], **Sewunet Ademe**[1], **Tarikuwa Natnael**[2], **Gete Berihun**[2], **Masresha Abebe**[2], **Yeshiwork Alemnew**[3], **Alemebante Mulu**[4], **Yordanos Mezemir**[5], **Abayneh Melaku**[6], **Taffere Addis**[6], **Emaway Belay**[7], **Zebader Walle**[7], **Lake Kumlachew**[8], **Abraham Teym**[8], **Metadel Adane**[2]*

1 Department of Nursing, School of Nursing and Midwifery, College of Medicine and Health Sciences, Wollo University, Dessie, Ethiopia, 2 Department of Environmental Health, College of Medicine and Health Sciences, Wollo University, Dessie, Ethiopia, 3 Department of Biology, College of Natural Sciences, Wollo University, Dessie, Ethiopia, 4 Faculty of Informatics, Department of Computer Science, St. Mary's University, Addis Ababa, Ethiopia, 5 Debre Birhan Health Science College, Debre Birhan, Ethiopia, 6 Ethiopian Institute of Water Resources (EIWR), Addis Ababa University, Addis Ababa, Ethiopia, 7 Department of Public Health, College of Health Sciences, Debre Tabor University, Debre Tabor, Ethiopia, 8 Department of Environmental Health, Health Sciences College, Debre Markos University, Debre Markos, Ethiopia

☯ These authors contributed equally to this work.
* metadel.adane2@gmail.com

**Data Availability Statement:** All relevant data are within the manuscript and its Supporting Information files.

## Abstract

### Background

Cases of coronavirus disease (COVID-19) are increasing at an alarming rate throughout the world, including Ethiopia. Food handlers in food and drink establishments are at high risk of exposure to the virus due to their many daily contacts with customers. Since there is a paucity of evidence about infection prevention practices and associated factors among this high-risk group in Ethiopia including in Dessie City and Kombolcha Town, this study was designed to address this gap.

### Method

An institution-based cross-sectional study was conducted among 422 food handlers in Dessie City and Kombolcha Town food and drink establishments in July and August 2020. The study participants were selected using a simple random sampling technique. Data were collected by trained data collectors using a pretested structured questionnaire and an on-the-spot observational checklist. Data were entered into EpiData version 4.6 and exported to STATA version 14.0 for data cleaning and analysis. Data were analyzed using bivariable and multivariable logistic regression model at 95% confidence interval (CI). From the bivariable analysis, variables with a *p*-value <0.25 were retained into multivariable analysis. Finally, variables that had a *p*-value <0.05 were declared as factors significantly associated with good infection prevention practices of COVID-19 among food handlers.

**Funding:** Wollo University funded this study. The funders had no role in study design, data collection and analysis, decision to publish, or preparation of the manuscript.

**Competing interests:** The authors have declared that no competing interests exist.

**Abbreviations:** AOR, adjusted odds ratio; CI, confidence interval; COVID, corona virus disease; PPE, personal protective equipment; WHO, World Health Organization.

## Main findings

The overall rate of good practice in infection prevention among food handlers was 43.9% (95% CI: 39.2–48.4%). Among the total 401 food handlers, 79.8% had good knowledge and 58.4% had a favorable attitude about COVID-19 infection prevention. Factors significantly associated with good COVID-19 infection prevention practices were: educational status of college or above (AOR = 1.97; 95% CI: 1.32–3.75), food handling work experience greater than five years (AOR = 2.55; 95% CI: 1.43–5.77), availability of written guidelines within the food and drink establishment (AOR = 2.68; 95% CI: 1.52–4.75), and taking training about infection prevention (AOR = 3.26; 95% CI: 1.61–6.61).

## Conclusion

Our findings showed that around one-third of food handlers had good infection prevention practices. Thus, to reduce COVID-19 transmission, integrated work is urgently needed to further improve food handlers' good practices, knowledge and attitude about infection prevention through providing health education, training and by making written infection prevention guidelines available in food and drink establishments.

## Introduction

Around the world, coronavirus disease (COVID-19) is the most overwhelming problem of the first part of the 21$^{st}$ century. Caused by severe acute respiratory syndrome, coronavirus 2 (SARS-CoV-2) was first reported in Wuhan, China in December 2019 and then rapidly spread throughout countries and territories outside of China [1, 2]. COVID-19 results in morbidity and mortality ranging from mild respiratory illness to severe acute respiratory distress syndrome, septic shock and other metabolic and homeostasis disorders and death [3]. Even though COVID-19 affects the whole population, the most frequent occurrence of fatal acute respiratory distress syndrome (ARDS) has been in older adults and people who have existing chronic medical conditions such as diabetes, cancer, hypertension and diseases of the heart, lung and kidneys [4–6]. COVID-19 exacerbates the existing social, political, religious and socio-economic crises in the whole population [7, 8].

According to a World Health Organization (WHO) report, 30.6 million COVID-19 cases with 950,000 deaths worldwide had been recorded by September 20, 2020. On that date, Africa accounted for 1,145,397 COVID-19 cases with 24,757 deaths, while in Ethiopia the number of COVID-19 cases reached 68,131 and confirmed deaths reached 1,089 [9].

SARS-CoV-2 appears to be transmitted from person to person through direct contact with respiratory droplets, saliva, discharge from the nose when the infected person coughs or sneezes, or indirectly through contaminated objects and surfaces as previously seen in SARS-CoV and MERS-CoV, the two other zoonotic coronaviruses [10, 11]. Implementation of stringent infection prevention measures is the most effective way to reduce the spread of the COVID-19 [11]. WHO has strongly recommended that community members, especially those working in crowded areas such as food and drink establishments, cover their noses and mouths with a tissue or elbow when coughing and sneezing, wear a mask, keep social distancing, wash their hands with soap or use appropriate alcohol-based hand rub and use appropriate personal protective equipment (PPE), all of which are simple and low-cost protective

procedures against COVID-19 transmission as compared to the serious impact of dealing with this disease [12, 13].

Many factors can affect food handlers' practice of coronavirus infection prevention strategies, such as availability of supplies, knowledge, attitude, training and socio-demographic characteristics. Hence, to overcome the problem of COVID-19, all food workers should follow physical distancing guidance of at least 2 meters (6 feet), wear a face mask and clean all surfaces with which employees and customers come into contact [14].

Failure to follow proper infection prevention strategies puts communities at risk. According to existing reports, despite the increment of COVID-19 prevalence in many developed and developing countries, the practice of infection prevention strategies among community members is not well studied [15]. Eastern Amhara, which includes the urban areas of Dessie City and Kombolcha Town, is one of the hotspot areas for COVID-19 due to the nearby road running from Djibouti to Afar to Batti City to Kombolcha Town and then to Dessie City. Residents and travellers in Dessie City and Kombolcha Town use food and drink establishments together, increasing the transmission of COVID-19. Therefore, this study was designed to assess COVID-19 infection prevention practices and associated factors among food handlers in Dessie City and Kombolcha Town food and drink establishments, aiming to provide key information to guide policy makers about the problem while they are designing intervention guidelines to improve COVID-19 infection prevention practices.

## Methods and materials

### Study area description

The study was carried out in Dessie City and Kombolcha Town, both found in South Wollo Zone, one of the 13 zones within the Amhara Regional State, Ethiopia. Dessie City and Kombolcha Town are located 401 km and 377 km from Addis Ababa with an average elevation of 2510 m and 1857 m above sea level, respectively. The 2007 national census conducted by the Central Statistical Agency) [16] of Ethiopia reported that Dessie City had a total population of 151,174, of which 72,932 were male and 78,242 were female, whereas Kombolcha Town had 126,144 total population, 60,226 male and 65,918 female. According to a 2020 Ethiopia Ministry of Urban Development and Construction report, there were 73 manufacturing companies, 400 service trades, six fuel stations, nine banks and one microfinance organization in Dessie City. Kombolcha Town had 3 manufacturing companies, 45 wholesale and 1,780 retail trades, 21 garages, 6 fuel stations, three banks and two microfinance organizations.

### Study design, period and population

An institution-based cross-sectional study was conducted in July and August 2020 among food handlers in food and drink establishments in Dessie City and Kombolcha Town. The study population was food handlers among the selected food and drink establishments. Food and drink establishments that had no eligible food handlers during the data collection period were excluded.

### Sample size determination and sampling techniques

The sample size was determined using a single population proportion formula with the assumptions of expected good COVID-19 infection prevention practices among food handlers of 50% due to the absence of previous studies in a similar setting, $Z_{\alpha/2}$ value 1.96 at 95% confidence interval (CI) and 5% margin of error. After considering a 10% non-response rate from the calculated sample size of 384, the final sample size was 422.

The total sample size was proportionally allocated based on the total of 349 and 204 food and drink establishments (including hotels, restaurants, bars/restaurants, cafeterias and butcher houses) in Dessie City and Kombolcha Town, respectively. For each town, the proportionate number of study subjects was determined using, $n = nf/N * ni$ where, $ni$ = number of food and drink establishments in each town, $nf$ = total sample size, $N$ = total number of food and drink establishments in Dessie City and Kombolcha Town all together. Therefore, the numbers of food and drink establishments in the two towns by proportional allocation were 266 from Dessie City and 156 from Kombolcha Town.

The sampling units were food and drink establishments that had at least one food handler whose age was 18 years or above. The sampling frame and source population were prepared by creating a list of food and drink establishments that had at least one food handler, aged 18 years or above. Then, those eligible food and drink establishments were coded to easily differentiate them during establishment-to-establishment visits at the time of data collection. Using the sampling frame, a systematic random sampling with a fixed interval of 'K' was used to get the next food and drink establishment within each town. To select the food and drink establishment paired with one eligible study participant, the data collector started with a bench mark of the known location and then walked straight forward to identify each food and drink establishment.

Further, when more than one eligible study participant was present in a selected food and drink establishment, a lottery method was used to select one study participant to estimate the proportion of good or poor practices of infection prevention among the study participants relative to the total sample size. To minimize the non-response rate, if one eligible study participant was not available from a selected food and drink establishment, a second visit was made the same day. If they were again not available, another visit was made the next day. If not available after the third visit, that participant was taken as non-respondent.

## Measurement of the outcome variable

The outcome variable of this study was the good or poor practices of COVID-19 infection prevention among food handlers. Nine yes/no questions, one observational checklist and five multiple choice infection prevention practices questions were asked with a minimum score of 1 and maximum score of 25. Good infection prevention practice (the variable of interest) was determined for food handlers who scored 75% or above, whereas poor infection prevention practices refers to those food handlers who scored below 75% on the practice questions [17].

## Operational definitions

**Infection prevention** is a scientific approach and practical solution designed to place barriers between a susceptible host and the microorganism [18].

**COVID-19** is a disease caused by SARS-CoV-2, a new member of a large family of transmissible viruses [19].

**Food handlers** are any persons who directly handle packaged or unpackaged food, food equipment and utensils, or food contact surfaces and are therefore expected to comply with food hygiene requirements [20].

**Food and drink establishments** are institutions that provide food and drink services to a relatively large number of users [20], which in this study included hotels, restaurants, bars/restaurants, cafeterias and butcher houses.

**Good knowledge** about COVID-19 infection prevention practices refers to study participants who correctly answered more than or equal to 70% of knowledge questions [21].

**Poor knowledge** about COVID-19 infection prevention practices refers to study participants who correctly answered less than 70% of knowledge questions [21].

**Favorable attitude** towards COVID-19 infection prevention practices refers to study participants who scored greater than or equal to the mean score on attitude questions [22].

**Unfavorable attitude** towards COVID-19 infection prevention practices refers to study participants who scored less than the mean score on attitude questions [22].

## Data collection and quality control

Data were collected using a structured, pre-tested questionnaire and an on-the-spot observational checklist. The questionnaire and observational checklist were adapted from a variety of reviewed literature, such as WHO and EMOH guidelines about COVID-19 infection prevention [7, 13, 23, 24]. The prepared English version of the questionnaire (S1 Questionnaires) was translated to Amharic (S2 Questionnaires), which is a local language, and then retranslated back to English by bilingual experts to ensure its consistency.

Independent variables were socio-demographic characteristics, availability of supplies (10 items on observational checklist and 9 interview questions), knowledge about and attitude towards COVID-19 infection prevention practices. Eleven yes/no questions were used to assess food handlers' knowledge, with the minimum and maximum scores being 2 and 11, respectively, and the mean score 9.33 (SD [standard deviation]: ±1.97). In order to determine food handlers' attitude regarding COVID-19 infection prevention practices, nine attitude statements were given with the value of each option being strongly disagree 1, disagree 2, neutral 3, agree 4 and strongly agree 5. The minimum and maximum scores were 9 and 36, respectively, with a mean score of 27.59 and SD of ±3.81.

Four data collectors and two supervisors were recruited who were BSc. nurses and environmental health professionals, respectively. All data collectors and supervisors had previous experience of COVID-19 data collection and supervision activities in a similar setting. Two days of training were given to data collectors and supervisors by the principal investigator with respect to the aim of the study, data collection procedures, contents of the structured questionnaire and observational checklist, including how to record data, how to keep social distance and wear a mask during an interview and the ethical aspects of approaching the participants. Data collectors kept a social distance of at least 2 meters, wore masks and approached the participants politely and respectfully at the time of the interview. The supervisors monitored the data collection process daily; if a problem arose data collectors tried to solve it or they contacted the principal investigator by mobile or in person.

To assure the data quality, the questionnaire and observational checklist were pre-tested among a number of food handlers 10% of the sample size at Haike Town food and drink establishments, an area not included in this study. Based on the pre-test result, amendments of unclear and vague questions were made. Using the pre-test result, internal reliability was checked, which gave Cronbach's Alpha value of 0.920, 0.729 and 0.813 for practices, knowledge and attitude questions, respectively. Content validity of the instrument was evaluated by one senior environmental health professional and one nursing educator. Each questionnaire and observational checklist was checked daily for completeness; any incomplete questionnaire or checklist was corrected the same day by visiting the food and drink establishments again.

## Data management and analysis

The collected data were coded and entered into EpiData version 4.6 and exported to STATA version 14.0 for data cleaning and analysis. Means with standard deviation (SD) were reported

for continuous variables and frequencies with proportion were computed for categorical variables.

Data were analyzed using a binary logistic regression model at 95% confidence interval and variables with $p$-value< 0.25 during the bivariable analysis were entered into a multivariable logistic regression analysis to control confounding variables. Adjusted odds ratio (AOR) with 95% CI was calculated to determine the strength of association; variables with a $p$-value less than 0.05 were declared as statistically significant. The **Hosmer-Lemeshow test** ($p$-value = 0.910) showed the model was fitted. Multicollinearity was tested using the Variance Inflation Factor (VIF) and tolerance test. Since VIF = 1.516, which is less than 5, and tolerance value = 0.667, which is greater than 0.2, it showed that there was no multicollinearity.

## Ethical considerations

Ethical clearance was obtained from the ethical review committee of Wollo University, College of Medicine and Health Sciences (Protocol number: CMHS/451/013/2020). After a request for cooperation from the health bureau of Dessie City and Kombolcha Town, permission was obtained to conduct the study. Written consent was obtained from study subjects prior to data collection. Data collectors wore facemasks to prevent COVID-19 transmission to or from food handlers. Furthermore, social distancing was maintained between data collectors and food handlers based on WHO recommended guidelines. Participants were informed of the objective of the study and assured that confidentiality would be kept. Data collection was conducted anonymously.

## Results

### Socio-demographic characteristics of participants

In this study, out of the total 422 eligible food handlers, 401participated, giving a response rate of 95%. Two hundred seventeen (54.1%) of the participants were female. More than one-third (41.9%) of food handlers were married, whereas just over half (51.1%) were single. The age of study participants ranged from 18 to 65 years with a median age of 26 years (IQR [interquartile range] = 23–32) and 273 (68.1%) under age 30. About 207 (51.7%) food handlers were waiters in the food and drink establishments. Nearly half (49.4%) of food and drink establishments had 5–10 food handlers and 181 (45.2%) food handlers had a monthly income of $27–$54 US (Table 1).

### Availability of supplies

Our results showed that about 96 (23.9%) of the food and drink establishments had written COVID-19 prevention guidelines, 43 (10.7%) had a COVID-19 infection prevention focal person, 166 (41.4%) had a specific budget for PPE and 222 (55.4%) had personal protective equipment against COVID-19 for their workers. The majority 362 (90.3%) of food and drink establishments had arranged COVID-19 infection prevention training for food handlers (Table 2).

### Knowledge about COVID-19 infection prevention strategies

To determine the level of food handlers' knowledge about COVID-19 infection prevention strategies, 11 questions that had yes/no alternatives were asked in the interviews. Our results showed that around three-fourths (79.8%) (95% CI: 75.183.3%) of food handlers had good knowledge about infection prevention (Table 3). The majority of food handlers 371 (92.5%) knew that all people are at risk of COVID-19 including themselves. Almost as many 358

**Table 1. Socio-demographic characteristics and bivariable analysis with infection prevention practices among food handlers at food and drink establishments in Dessie City and Kombolcha Town, Northeastern Ethiopia, July–August 2020.**

| Variables | Category | Frequency (N = 401) | Infection prevention practices status | | COR (95% CI) | p-value |
|---|---|---|---|---|---|---|
| | | | Good | Poor | | |
| | | n(%) | n | n | | |
| Age (years) | 18–30 | 273(68.1) | 95 | 178 | Ref | |
| | 31–49 | 114(28.4) | 71 | 43 | 3.09(1.97–4.87) | <0.001 |
| | > = 50 | 14(3.5) | 10 | 4 | 4.68(1.43–15.33) | 0.011 |
| Sex | Male | 184(45.9) | 85 | 99 | 1.19(0.80–176) | 0.392 |
| | Female | 217(54.1) | 91 | 126 | Ref | |
| Marital status | Married | 168(41.9) | 89 | 79 | Ref | |
| | Single | 205(51.1) | 75 | 130 | 0.51(0.34–0.78) | 0.252 |
| | Divorced | 17(4.2) | 6 | 11 | 0.48(0.17–1.37) | 0.372 |
| | Widowed | 11(2.8) | 6 | 5 | 1.07(0.31–3.63) | 0.920 |
| Educational status | Unable to read and write | 19(4.7) | 12 | 7 | Ref | |
| | Informal education | 19(4.7) | 9 | 10 | 1.54(0.75–2.66) | 0.330 |
| | Primary | 65(16.2) | 27 | 38 | 1.80(0.93–2.71) | 0.102 |
| | Secondary | 207(51.7) | 68 | 139 | 2.64(1.42–5.03) | 0.012 |
| | College or above | 91(22.7) | 60 | 31 | 2.07(1.52–3.95) | 0.003 |
| Job position in the food and drink establishment | Cook | 101(25.2) | 51 | 50 | Ref | |
| | Dish washer | 62(15.5) | 30 | 32 | 2.39(1.23–4.65) | 0.767 |
| | Waiter | 207(51.7) | 43 | 164 | 2.12(1.12–4.03) | 0.310 |
| | Other | 31(7.8) | 18 | 13 | 2.79(1.74–4.47) | 0.650 |
| Number of food handlers in each food and drink establishment (persons) | <5 | 85(21.2) | 35 | 50 | Ref | |
| | 5–10 | 198(49.4) | 69 | 129 | 0.76(0.45–1.29) | 0.312 |
| | 11–15 | 63(15.7) | 32 | 31 | 1.48(0.77–2.84) | 0.251 |
| | >15 | 55(13.7) | 40 | 15 | 3.81(1.83–7.94) | 0.302 |
| Years of service | <1 | 157(39.2) | 53 | 104 | 0.66(0.43–1.03) | 0.267 |
| | 1–5 | 196(48.8) | 85 | 111 | Ref | |
| | > 5 | 48(12.0) | 38 | 10 | 4.96(2.34–10.52) | < 0.001 |
| Food handlers' monthly salary (USD [United States Dollars], $)* | <28 | 86(21.4) | 13 | 73 | Ref | |
| | 28–56 | 181(45.2) | 79 | 102 | 4.35(2.25–8.41) | 0.251 |
| | > 56 | 134(33.4 | 84 | 50 | 9.43(4.75–18.73) | 0.410 |

*Average exchange rate of 1$US to Ethiopian Birr (ETB) in July and August 2020 was 1$US = 35.6395 ETB.

Other: Butcher, Kitchen manager and Vegetable station worker.

COR, crude odds ratio; Ref, reference category

(89.3%) food handlers knew that touching the nose, mouth and eyes is not recommended even if gloves are worn when giving service. Three hundred seventy-eight (94.3%) food handlers knew that washing hands frequently with water and soap and applying sanitizer would physically remove, inhibit or kill the coronavirus. Three hundred seventy-five (92.8%) food handlers knew the importance of social distance and 371 (92.5%) wore PPE at work (Table 4).

## Attitude towards COVID-19 infection prevention practices

Our findings showed that 234 (58.4%) (95% CI: 53.6–63.6%) food handlers had a favorable attitude about infection prevention practices (Table 3). More than three-quarters (79.3%) of

**Table 2.** Availability of supplies and its bivariable analysis with infection prevention practices among food handlers at food and drink establishments in Dessie City and Kombolcha Town, Northeastern Ethiopia, July–August 2020.

| Variables | Response | Frequency (N = 401) | Infection prevention practices status | | COR (95% CI) | p-value |
|---|---|---|---|---|---|---|
| | | | Good | Poor | | |
| | | n(%) | n | n | | |
| Availability of COVID 19 prevention guidelines in food and drink establishment (observation) | No | 305(76.1) | 110 | 195 | Ref | |
| | Yes | 96(23.9) | 66 | 30 | 3.90(2.39–6.37) | <0.001 |
| Availability of COVID 19 infection prevention focal person | No | 358(89.3) | 142 | 216 | Ref | |
| | Yes | 43(10.7) | 34 | 9 | 5.75(2.68–12.35) | 0.010 |
| Availability of specific budget for PPE in this COVID-19 era | No | 235(58.6) | 74 | 161 | Ref | |
| | Yes | 166(41.4) | 102 | 64 | 3.47(2.29–5.26) | 0.011 |
| Availability of personal protective equipment (observation) | No | 179(44.6) | 65 | 114 | Ref | |
| | Yes | 222(55.4) | 111 | 111 | 1.75(1.17–2.62) | 0.006 |
| COVID-19 infection prevention training arranged for food handlers | No | 39(9.7) | 13 | 26 | Ref | |
| | Yes | 362(90.3) | 163 | 199 | 1.64(0.82–3.29) | 0.265 |
| Posted information related to COVID-19 including emergency phone numbers visible on site (observation) | No | 275(68.6) | 87 | 188 | Ref | |
| | Yes | 126(31.4) | 89 | 37 | 3.20(3.28–8.23) | 0.265 |
| Services provided in accordance with COVID-19 safety measures | No | 80(20.0) | 29 | 51 | Ref | |
| | Yes | 321(80.0) | 147 | 174 | 1.49(0.90–2.46) | 0.456 |
| Availability of registration book for documenting events related to COVID-19 (observation) | No | 346(86.3) | 139 | 207 | Ref | |
| | Yes | 55(13.7) | 37 | 18 | 3.06(1.68–5.60) | 0.650 |
| Availability of visible posted order for waiters/servers and customers to practice social distancing and to avoid touching each other (observation) | No | 91(22.7) | 25 | 66 | Ref | |
| | Yes | 310(77.3) | 151 | 159 | 2.51(1.50–4.18) | 0.901 |
| Frequently washing and immersing tablecloths in a mixture of one part bleach and nine parts water for 10 minutes and finally rinsing with pure water | No | 286(71.3) | 108 | 178 | Ref | |
| | Yes | 115(28.7) | 68 | 47 | 2.39(1.53–3.71) | 0.451 |
| Having well-cleaned and ventilated service rooms, toilets, meeting halls and corridors (observation) | No | 128(31.9) | 46 | 82 | Ref | |
| | Yes | 273(68.1) | 130 | 143 | 1.62(1.05–2.50) | 0.329 |
| Daily cleaning of doors, walls, windows, tables, chairs and mobile phones using sanitizer or a solution containing one part bleach and nine parts water | No | 127(31.7) | 60 | 67 | Ref | |
| | Yes | 274(68.3) | 116 | 158 | 0.82(0.54–1.25) | 0.357 |
| Treating materials and equipment used by customers with a solution containing one part bleach and nine parts water | No | 273(68.1) | 96 | 177 | Ref | |
| | Yes | 128(31.9) | 80 | 48 | 3.10(1.99–4.75) | 0.452 |
| Rinsing brooms, brushes and utility gloves with bleach | No | 152(37.9) | 59 | 93 | Ref | |
| | Yes | 249(62.1) | 117 | 132 | 1.40(0.93–2.11) | 0.311 |
| Chairs are arranged 2 meters apart (observation) | No | 193(48.1) | 61 | 132 | Ref | |
| | Yes | 208(51.9) | 115 | 93 | 2.68(1.78–4.03) | 0.271 |

(*Continued*)

**Table 2.** (Continued)

| Variables | Response | Frequency (N = 401) n(%) | Infection prevention practices status | | COR (95% CI) | p-value |
|---|---|---|---|---|---|---|
| | | | **Good** n | **Poor** n | | |
| Accessibility of covered dust bins in each room (observation) | No | 81(20.2) | 30 | 51 | Ref | |
| | Yes | 320(79.8) | 146 | 174 | 1.43(0.86–2.36) | 0.264 |
| Availability of segregating materials to separate dry and liquid waste (observation) | No | 69(17.2) | 34 | 35 | Ref | |
| | Yes | 332(82.8) | 142 | 190 | 0.77(0.46–1.29) | 0.323 |
| Collecting and disposing of wastes properly (observation) | No | 63(15.7) | 29 | 34 | Ref | |
| | Yes | 338(84.3) | 147 | 191 | 0.90(0.53–1.55) | 0.709 |
| Ever taken COVID-19 infection prevention training | No | 343(85.5) | 132 | 211 | Ref | |
| | Yes | 58(14.5) | 44 | 14 | 5.02(2.65–9.52) | <0.001 |

COR, crude odds ratio; Ref, reference category.

study participants agreed that washing hands with soap or an alcohol-based antiseptic decreases the risk of transmission of COVID-19 while only 17 (4.2%) respondents strongly disagreed. Regarding social distancing, 303 (75.6%) food handlers agreed it was an important way to reduce the transmission of the coronavirus (Table 5).

## Food handlers' COVID-19 infection prevention practices

We found that 176 (43.9%) (95% CI: 39.2–48.4%) food handlers had good COVID-19 infection prevention practices. Among all respondents, 365 (91.0%) reported that they washed their hands regularly; of this group 131 (35.9%) demonstrated practical handwashing techniques perfectly. Among the remaining 36 (9.0%) respondents who did not wash their hands regularly, the unavailability of handwashing materials was the important reason. Three hundred eighty-nine (97%) food handlers reported that when they sneezed, they covered their nose and mouth with an elbow. Around 365 (91.0%) food handlers wore PPE, while the remaining 36 (9.0%) respondents did not wear PPE. Among the 36 (9.0%) food handlers who did not wear PPE, 10 (27.8%) thought that wearing PPE was not always necessary to prevent COVID-19. Masks were worn by 351 (87.8%) food handlers (Table 6).

**Table 3. Knowledge and attitude of food handlers and bivariable analysis with infection prevention practices among food handlers at food and drink establishments in Dessie City and Kombolcha Town, Northeastern Ethiopia, July–August 2020.**

| Variables | Response | Frequency (N = 401) n(%) | Infection prevention practices status | | COR (95% CI) | P–value |
|---|---|---|---|---|---|---|
| | | | **Good** n | **Poor** n | | |
| Knowledge | Good | 320(79.8) | 143 | 177 | 1.18(0.72–1.93) | 0.523 |
| | Poor | 81(20.2) | 33 | 48 | Ref | |
| Attitude | Favorable | 234(58.0) | 95 | 139 | 1.38(0.92–2.06) | 0.116 |
| | Unfavourable | 167(42.0) | 81 | 86 | Ref | |

COR, crude odds ratio; Ref, reference category

**Table 4. Frequency and percentage distribution of knowledge about COVID-19 infection prevention practices among food handlers at food and drink establishments in Dessie City and Kombolcha Town, Northeastern Ethiopia, July–August 2020.**

| Variables | Responses | Frequency (N = 401) | Percentage (%) |
|---|---|---|---|
| All microorganisms including coronaviruses are removed by washing with water and antiseptic agents. | No | 42 | 10.5 |
| | Yes | 359 | 89.5 |
| All people are at risk of COVID-19. | No | 30 | 7.5 |
| | Yes | 371 | 92.5 |
| Washing hands with soap and water and applying sanitizer would inhibit or kill the coronavirus. | No | 23 | 5.7 |
| | Yes | 378 | 94.3 |
| There is a need to wash hands before and after touching things. | No | 27 | 6.7 |
| | Yes | 374 | 93.3 |
| Staying 2 meters apart from other individuals prevents the transmission of COVID-19. | No | 29 | 7.2 |
| | Yes | 372 | 92.8 |
| The incubation period of coronavirus is 14 days. | No | 34 | 8.5 |
| | Yes | 367 | 91.5 |
| Wearing PPE (such as mask, goggle, gloves) decreases the risk of transmission of COVID-19. | No | 30 | 7.5 |
| | Yes | 371 | 92.5 |
| A cloth mask can be reused after washing with soap and water, decontaminating and drying but a surgical mask cannot. | No | 30 | 7.5 |
| | Yes | 371 | 92.5 |
| Wearing gloves does not replace the need for handwashing or use of antiseptic hand rubs. | No | 42 | 10.5 |
| | Yes | 359 | 89.5 |
| It is not recommended to touch nose, mouth, eyes even if you don glove when you give service. | No | 43 | 10.7 |
| | Yes | 358 | 89.3 |
| Surgical gloves can be reused. | No | 340 | 84.8 |
| | Yes | 61 | 15.2 |

**Table 5. Frequency and percentage distribution of attitude towards COVID-19 infection prevention practices among food handlers at food and drink establishments in Dessie City and Kombolcha Town, Northeastern Ethiopia, July–August 2020.**

| Attitude variables | Strongly Disagree n(%) | Disagree n(%) | Neutral Agree n(%) | Agree n(%) | Strongly Agree n(%) |
|---|---|---|---|---|---|
| Washing hands with soap or an alcohol-based antiseptic decreases the risk of transmission of COVID-19. | 17(4.2) | 16(4.0) | 12(3.0) | 318 (79.3) | 38(9.5) |
| Gloves and mask provide complete protection against COVID-19. | 18(4.5) | 212 (52.9) | 65(16.2) | 93(23.2) | 13(3.2) |
| Handwashing is unnecessary when gloves are worn | 29(7.2) | 263 (65.6) | 64(16.0) | 44(11.0) | 1(0.2) |
| Frequent handwashing damages skin and causes cracking, dryness, irritation and dermatitis. | 16(4.0) | 115 (28.7) | 62(15.5) | 205 (51.1) | 3(0.7) |
| You have a very low risk of acquiring COVID-19 from others. | 90(22.5) | 259 (64.6) | 13(3.2) | 34(8.5) | 5(1.2) |
| COVID-19 is like the common cold, which has no serious effect. | 158(39.4) | 189 (47.1) | 12(3.0) | 30(7.5) | 12(3.0) |
| Gloving is a useful strategy for reducing risk of transmission of novel coronavirus. | 11(2.8) | 21(5.2) | 21(5.2) | 336 (83.8) | 12(3.0) |
| Social distancing is a basic technique to reduce the transmission of novel coronavirus. | 13(3.2) | 13(3.2) | 16(4.0) | 303 (75.6) | 56(14.0) |
| Being locked down prevents the transmission of novel coronavirus. | 23(5.7) | 21(5.2) | 16(4.0) | 226 (56.4) | 115(28.7) |

**Table 6. Frequency and percentage distribution of infection prevention practices related to COVID-19 among food handlers at food and drink establishments in Dessie City and Kombolcha Town, Northeastern Ethiopia, July–August 2020.**

| Variables | Responses | Frequency (N = 401) | Percentage (%) |
|---|---|---|---|
| Washing hands regularly (N = 401) | No | 36 | 9.0 |
| | Yes | 365 | 91.0 |
| Occasions to wash hands (N = 365)* | Before contact with things | 183 | 50.1 |
| | After contact with things | 311 | 85.2 |
| | Before preparing food | 203 | 55.6 |
| | After preparing food | 125 | 34.2 |
| | If I look or feel dirty | 134 | 36.7 |
| | Before leaving home | 79 | 21.6 |
| | Before entering the home from outside | 229 | 62.7 |
| | Before going to the toilet | 88 | 24.1 |
| | After going to the toilet | 160 | 43.8 |
| | Before donning gloves | 52 | 14.2 |
| | After removing gloves | 26 | 7.1 |
| | After sneezing | 18 | 4.9 |
| Observing food handlers' method of washing their hands (N = 365) | Demonstrate imperfect method | 234 | 64.1 |
| | Demonstrate perfect method | 131 | 35.9 |
| Materials used for handwashing (N = 365)* | With water only | 100 | 27.4 |
| | With plain soap and water | 294 | 80.5 |
| | With anti-bacterial soap and water | 133 | 36.4 |
| | With alcohol/sanitizer | 219 | 60.0 |
| Reason not to wash hands (N = 36) | Have no information how to wash hand | 8 | 22.2 |
| | Unavailability of hand washing materials | 21 | 58.3 |
| | Negligence | 4 | 11.1 |
| | Other | 3 | 8.4 |
| Using antiseptic hand rub(N = 401) | No | 26 | 6.5 |
| | Yes | 375 | 93.5 |
| Wearing personal protective equipment to prevent COVID-19(N = 401) | No | 36 | 9.0 |
| | Yes | 365 | 91.0 |
| Type of PPE used (N = 365)* | Gloves | 145 | 39.7 |
| | Gown | 49 | 13.4 |
| | Cap | 43 | 11.8 |
| | Goggles | 17 | 4.7 |
| | Mask | 351 | 87.8 |
| Reason not to use personal protective equipment (N = 36) | Lack of materials | 8 | 22.2 |
| | Lack of awareness | 3 | 8.3 |
| | Difficult to work with PPE | 7 | 19.4 |
| | Not always necessary | 10 | 27.8 |
| | Carelessness | 3 | 8.4 |
| | Other | 5 | 13.9 |
| Sneezing by covering with elbow (N = 401) | No | 12 | 3.0 |
| | Yes | 389 | 97.0 |
| Touching nose, mouth and/or eyes with hand when hosting customers (N = 401) | No | 333 | 83.0 |
| | Yes | 68 | 17.0 |
| Keeping social distance of 1 meter apart (N = 401) | No | 65 | 16.2 |
| | Yes | 336 | 83.8 |

(*Continued*)

**Table 6.** (Continued)

| Variables | Responses | Frequency (N = 401) | Percentage (%) |
|---|---|---|---|
| Taking shower and changing clothes before contact with family (N = 401) | No | 241 | 60.1 |
| | Yes | 160 | 39.9 |
| Using infection prevention guidelines for COVID-19 in working area (N = 401) | No | 287 | 71.6 |
| | Yes | 114 | 28.4 |
| Ever taken training about IP of COVID-19 (N = 401) | No | 343 | 85.5 |
| | Yes | 58 | 14.5 |

NB: Frequencies and percentages do not add up to give n* values or100% due to multiple responses

### Bivariable and multivariable logistic regression analysis

From the bivariable analysis; age, educational status, years of experience, availability of COVID-19 prevention guidelines, availability of COVID-19 infection prevention focal person, availability of specific budget for COVID-19 infection prevention, availability of PPE in food and drink establishments, ever having taken COVID-19 infection prevention training and attitude toward COVID-19 infection prevention had a *p*-value <0.25 and were retained into the multivariable analysis.

From the multivariable analysis, respondents who had an educational status of college or above were almost twice as likely to have good practices than those who were unable to read and write (AOR = 1.97; 95% CI:1.32–3.75). Food handlers who had work experience of greater than 5 years were almost 2.55 times (AOR = 2.55; CI: 1.43–5.77) more likely to have good COVID-19 infection prevention practices than those who had 5 or fewer years' experience of handling food. Those food handlers who had written COVID-19 prevention guidelines in their workplace were almost 2.68 times more likely to practice COVID-19 infection prevention strategies than those who did not (AOR = 2.68; 95% CI: 1.52–4.75). Food handlers who had taken COVID-19 infection prevention training were almost 3.26 times more likely to have good COVID-19 infection prevention practices than those who had not taken such training (AOR = 3.26; 95% CI: 1.61–6.61) (Table 7).

## Discussion

We conducted a cross-sectional study in Northeastern Ethiopia to assess the level of COVID-19 infection prevention practices among food handlers in Dessie City and Kombolcha Town food and drink establishments. Out of 401 food handlers, around three-fourths had good knowledge about COVID-19 infection prevention strategy practices. Regarding the attitude of food handlers about COVID-19 infection prevention strategy practices, 58.4% had a favorable attitude. We found that educational status, years of experience, availability of COVID-19 infection prevention guidelines and ever having taken COVID-19 infection prevention training had a statistically significant association with the food handlers' practices of COVID-19 infection prevention strategies.

In this study, 43.9% of food handlers had good COVID-19 infection prevention practices. Our study finding was consistent with those of previous studies in Addis Zemen, Ethiopia (47.3%) [6] and Bangladesh (44.8%) [25]. This finding was lower than reported by studies in Amhara region (62%) [17], Uganda (74%) [26], China (89.7%) [27] and Pakistan (88.7%) [28]. The possible reasons for this discrepancy might include the difference in conduciveness to good practices of food handlers' working environment, availability of managerial support and educational level of study participants [17], difference in socio-demographic and

**Table 7. Multivariable analysis of factors associated with COVID-19 infection prevention practices among food handlers at food and drink establishments in Dessie City and Kombolcha Town, Northeastern Ethiopia, July–August 2020.**

| Variables | | Infection prevention practices status | | AOR (95% CI) | P-value |
|---|---|---|---|---|---|
| | | Good | Poor | | |
| | | n | n | | |
| Educational status | Unable to read and write | 7 | 12 | Ref | |
| | Informal education | 10 | 9 | 1.32(0.50–2.10) | 0.338 |
| | Primary school | 36 | 29 | 1.56(0.86–2.37) | 0.956 |
| | Secondary school | 141 | 66 | 2.40(1.22–4.73) | 0.543 |
| | College or above | 60 | 31 | 1.97(1.32–3.75) | 0.042 |
| Years of experience | <1 | 53 | 104 | 0.84(0.51–1.38) | 0.678 |
| | 1–5 | 85 | 111 | Ref | |
| | >5 | 38 | 10 | 2.55(1.43–5.77) | 0.025 |
| Availability of COVID-19 infection prevention guidelines in food and drink establishment | Yes | 66 | 30 | 2.68(1.52–4.75) | < 0.001 |
| | No | 110 | 195 | Ref | |
| Ever taken COVID-19 infection prevention training | Yes | 44 | 14 | 3.26(1.61–6.61) | < 0.001 |
| | No | 132 | 211 | Ref | |

Ref, reference category; AOR, adjusted odds ratio

environmental factors [26], difference in culture and socio economic status of respondents [27, 28]. On the other hand, the current study finding was higher than reported by studies conducted in Bale Zone, Ethiopia (36.8%) [29] and the Philippines (32.4%) [30]. The probable reasons for our higher finding are the differences in study participants where our study focused on food handlers, sample size and study period [29] and differences in socio-cultural practices, in the tool used for assessment of COVID-19 infection prevention practices, health policy and strategies to prevent COVID-19 in the two study localities and environmental factors [30].

Though the execution was questionable, most of the food handlers tried to follow Ethiopia's Ministry of Health and WHO infection prevention practices recommendations. These include regular wearing of PPE including a face mask, hand hygiene and social distancing. About 91.0% of food handlers reported wearing PPE, of whom 87.8% wore a face mask. Ninety-one percent and 83.8% of participants reported washing hands regularly and keeping social distance of one meter, respectively. These findings were supported by an Ethiopian study conducted at Jimma University Medical Center [31]. In contrast, a study conducted in Mizan-Aman Ethiopia revealed that about 81.8% of waiters did not wash their hands frequently [32].

The overall rate of good COVID-19 infection prevention knowledge was about 79.8%, in line with an online survey conducted in Ethiopia [33] and studies conducted in Jordan [34] and Nigeria [35]. This finding indicates the importance of enhancing COVID-19 knowledge through various techniques as a factor to improve the preventive strategies against the disease. Food handlers need to be encouraged to improve their attitude towards COVID-19 infection prevention. The current study's finding with respect to attitude is similar to that found in studies conducted in northern Ethiopia [36] and China [37].

This study identified factors significantly associated with COVID-19 infection prevention practices. Higher educational status was significantly associated with good practices of infection prevention among food handlers. This finding was supported by studies conducted in Addis Zemen, Ethiopia [6], Mizan-Aman, Ethiopia [32], Uganda [26], Iran [38] and China [39]. The probable reasons for this finding are that educated food handlers may have good

understanding, perception, knowledge and skill with respect to COVID-19 infection prevention measures; also they may easily access the recommendations of health care professionals about COVID-19 preventive measures.

Having long work experience was associated with good COVID-19 infection prevention practices among food handlers. This finding was supported by studies in Pakistan [28] and China [27]. The possible explanation is a difference in the understanding of the importance of COVID-19 infection prevention between inexperienced and more senior food handlers. As food handlers' work experience increases over time and they face challenges due to the impacts of poor infection prevention practices, their attitude about the benefit of infection prevention implementation improves. It is also possible that as workers' food handling skills increase, they find it easier to remember and perform the extra tasks required for infection prevention even when they are busy.

Availability of written COVID-19 guidelines increased good COVID-19 infection prevention practices. This finding was similar to that of studies in the Amhara region [17] and South Africa [40]. This could be because standardized COVID-19 prevention guidelines are used as the source of information about how to practices infection prevention strategies that reduce its transmission. Also, food handlers may become alarmed when they see COVID-19 prevention guidelines in the catering area and be motivated to become familiar with them.

The main purpose of COVID-19 infection prevention training was to demonstrate food and drink establishments' commitment to the well-being of customers and staff. In this study, food handlers who had taken COVID-19 infection prevention training were more likely to have good COVID-19 infection prevention practices than those who had not taken it. This result was supported by studies conducted in the Amhara region [17], Uganda [26] and China [41]. This might be because training can impact knowledge about the COVID-19, equip food handlers with information about COVID-19 infection prevention precautions and update or enhance skills in how to use PPE and implement infection prevention guidelines.

A weakness of our study may be that of social desirability bias during self-reporting; however, we tried to control social desirability bias by the employment of proxy (representative) data. Furthermore, the level of accuracy of the measuring instrument was revised after pre-testing the data collection tools. The findings of this study may not represent the situation at the national level, as the study was conducted only in Dessie City and Kombolcha Town food and drink establishments.

## Conclusion

This study revealed that more than one-third of food handlers had good infection prevention practices. Our results also showed that around three-fourths and more than half of food handlers had good knowledge and a favorable attitude about infection prevention, respectively. Factors significantly associated with good infection prevention practices included higher educational status, more years of experience, availability of WASH infection prevention guidelines and training. Due to the urgent need to control transmission of COVID-19, we recommend that integrated work be done to improve rates of good practices, knowledge and attitude about infection prevention among food handlers through providing these workers with health education and training. We also recommend that more studies be conducted on a larger scale, especially at regional and national levels.

## Supporting information

**S1 Questionnaires. English version of the questionnaires for COVID-19 infection prevention practices among food handlers in food and drink establishments of Dessie City and**

**Kombolcha Town, Northeastern Ethiopia.**
(DOCX)

**S2 Questionnaires. Amharic (local language) version of the questionnaires for COVID-19 infection prevention practices among food handlers in food and drink establishments of Dessie City and Kombolcha Town, Northeastern Ethiopia.**
(DOCX)

**S1 Dataset. Raw data for COVID-19 infection prevention practices among food handlers in food and drink establishments of Dessie City and Kombolcha Town, Northeastern Ethiopia.**
(XLSX)

## Acknowledgments

We thank Dessie City and Kombolcha Town Health Bureaus for the support we received during data collection. Our gratitude also goes to Dessie City and Kombolcha Town Culture and Tourism offices for providing the necessary data during the proposal development and data collection period. We also greatly appreciate the owners of Dessie City and Kombolcha Town food and drink establishments for their support and the information they provided when we needed it. We also extend our gratitude to food handlers, data collectors and supervisors for their valuable time and cooperation during the study. Last but not the least, we also thank the first English language speaker and Lisa Penttila for language editing of the manuscript.

## Author Contributions

**Conceptualization:** Atsedemariam Andualem, Metadel Adane.

**Data curation:** Atsedemariam Andualem, Metadel Adane.

**Formal analysis:** Atsedemariam Andualem, Metadel Adane.

**Funding acquisition:** Atsedemariam Andualem, Belachew Tegegne, Sewunet Ademe, Tarikuwa Natnael, Gete Berihun, Masresha Abebe, Yeshiwork Alemnew.

**Investigation:** Atsedemariam Andualem, Belachew Tegegne, Sewunet Ademe, Tarikuwa Natnael, Gete Berihun, Masresha Abebe, Yeshiwork Alemnew, Alemebante Mulu, Yordanos Mezemir, Abayneh Melaku, Taffere Addis, Emaway Belay, Zebader Walle, Lake Kumlachew, Abraham Teym, Metadel Adane.

**Methodology:** Atsedemariam Andualem, Belachew Tegegne, Sewunet Ademe, Tarikuwa Natnael, Gete Berihun, Masresha Abebe, Yeshiwork Alemnew, Alemebante Mulu, Yordanos Mezemir, Abayneh Melaku, Taffere Addis, Emaway Belay, Zebader Walle, Lake Kumlachew, Abraham Teym, Metadel Adane.

**Project administration:** Atsedemariam Andualem, Belachew Tegegne, Sewunet Ademe, Tarikuwa Natnael, Gete Berihun, Masresha Abebe, Yeshiwork Alemnew, Metadel Adane.

**Resources:** Atsedemariam Andualem, Belachew Tegegne, Sewunet Ademe, Tarikuwa Natnael, Gete Berihun, Masresha Abebe, Yeshiwork Alemnew, Alemebante Mulu, Yordanos Mezemir, Abayneh Melaku, Taffere Addis, Emaway Belay, Zebader Walle, Lake Kumlachew, Abraham Teym, Metadel Adane.

**Software:** Atsedemariam Andualem, Belachew Tegegne, Sewunet Ademe, Tarikuwa Natnael, Gete Berihun, Masresha Abebe, Yeshiwork Alemnew, Alemebante Mulu, Yordanos

Mezemir, Abayneh Melaku, Taffere Addis, Emaway Belay, Zebader Walle, Lake Kumla-chew, Abraham Teym, Metadel Adane.

**Supervision:** Atsedemariam Andualem, Belachew Tegegne, Sewunet Ademe, Tarikuwa Nat-nael, Gete Berihun, Masresha Abebe, Yeshiwork Alemnew, Alemebante Mulu, Yordanos Mezemir, Abayneh Melaku, Taffere Addis, Emaway Belay, Zebader Walle, Lake Kumla-chew, Abraham Teym, Metadel Adane.

**Validation:** Atsedemariam Andualem, Belachew Tegegne, Sewunet Ademe, Tarikuwa Nat-nael, Gete Berihun, Masresha Abebe, Yeshiwork Alemnew, Alemebante Mulu, Yordanos Mezemir, Abayneh Melaku, Taffere Addis, Emaway Belay, Zebader Walle, Lake Kumla-chew, Abraham Teym, Metadel Adane.

**Visualization:** Atsedemariam Andualem, Belachew Tegegne, Sewunet Ademe, Tarikuwa Nat-nael, Gete Berihun, Masresha Abebe, Yeshiwork Alemnew, Alemebante Mulu, Yordanos Mezemir, Abayneh Melaku, Taffere Addis, Emaway Belay, Zebader Walle, Lake Kumla-chew, Abraham Teym, Metadel Adane.

**Writing – original draft:** Atsedemariam Andualem, Metadel Adane.

**Writing – review & editing:** Atsedemariam Andualem, Metadel Adane.

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
