## [Decision Letter · Decision Letter 0]

22 Jun 2021

PONE-D-21-12395

COVID-19 Infection Prevention and Control Practices among Food Handlers in Food and Drinking Establishments of Dessie City and Kombolcha Towns in Northeastern Ethiopia

PLOS ONE

Dear Dr. Metadel Adane

Thank you for submitting your manuscript to PLOS ONE. After careful consideration, we feel that it has merit but does not fully meet PLOS ONE’s publication criteria as it currently stands. Therefore, we invite you to submit a revised version of the manuscript that addresses the points raised during the review process.

The findings of the study are important for program managers at the local municipal councils that regulate and control the food service establishments during COVID 19 pandemic. However, fundamental issues are indicated to be provided mainly for the methodology section (sample size calculation) and the statistical analysis.  The manuscript could be greatly strengthened by aggressive editing for the majority of the study sections.

Please note that your manuscript was reviewed by 11 experts in the field due to the sensitivity of the topic. There is consensus agreement that the idea of the article is interesting but also consensus agreement that the article required additional work. The reviewers identified many important problems and provided copious comments (enclosed).

The manuscript could be greatly strengthened by considering editing according to the specific Reviewers’ comments.

The reviewers had several comments. To allow you to easily understand and address them, the Academic Editor has summarized here the main concerns and suggestions that need to be addressed at revision (only the concerns summarized below need to be answered in your Response to Reviewers): 

Overall General comment:

Please note that further language improvements is indicated. Consider revising the spelling, grammar, diction, and syntax throughout the manuscript for increased clarity. The paragraphs should be made brief as there is a lot of repetition of words/phrases (refer to reviewer specific comments for some sentences as example). Consider replacing some words as described by the reviewers to be more specific and precise e.g : in the Abstract consider replacing "prevent" with "reduce" - "Thus, ensuring infection prevention and control practices for this high-risk group is urgently required to prevent COVID-19 transmission."

**Main points to be addressed:**

Abstract:

Consider adding one paragraph as background discussing the hypothesis of the study.Formulate the objective with 2 to 3 aimsDelete the non-related conclusion¨ Thus, ensuring infection prevention and control practices for this high-risk group is urgently required to prevent COVID-19 transmission}.Consider adding the study design and too to the methodology and remove the statistical tests description.Consider re-arranging the results Report proportions for knowledge, attitude and then finally practice. Then report the results for the multivariable logistic regression on factors associated with good practice. Although attitude was lower, it may not be related to poor practices. I think the authors might be better off suggesting how the uptake of good practices can be increased?Consider removing from the result of the abstract sentences that out of context. Delete ".... whereas the poor practice of infection prevention and control was 56.1% (95% CI: 51.4%-60.8%)" since the first part of the sentence already captures the main result.restrict to provide conclusions out of the study ¨Our findings showed that three-fourths of food handlers had good knowledge and more than half had a favorable attitude about infection prevention and control. Thus, integrated work is urgently needed to prevent COVID-19 transmission by further improving food handlers’ good practice of infection prevention and control}.Consider writing the keywords in an alphabetical order. 

Introduction:

the first 2- paragraphs should be merged with the following text added at the end of first paragraph while the rest of the text should be deleted as it is only a repetition: {According to a World Health Organization (WHO) report on 20 September 2020, 30.6 million positive cases with 950,000 deaths worldwide have been recorded. On that date, Africa accounted for 1,145,397 COVID-19 positives with a total death of 24,757; and in Ethiopia the number of COVID-19 cases reached 68,131 and confirmed deaths reached 1,089 [9].In the last paragraph the following sentence should be either deleted or re-written as it does not clearly interpret what is author trying to convey: {Implementation of IPC practices based on WHO and Ethiopian Ministry of Health (EMoH) guidelines is rapid, decisive, and collective action which saves millions of lives [13, 16, 17].}

Objectives

Consider formulating the objectives of the study to be more focused  as the given objective is too genialized. 

Methodology

Study Area:

It is not clear why only Dessie and Kombolcha town were included in the study.Add some description about the socioeconomic status of the area to generate some idea about the general people perceptions and attitudes in the study area.

Approach

•  A flowchart would add useful information to the reader of the paper. Particularly because not the entire food establishments were included.

Study design, period, and population:

consider mor clarification for the terminology : “The source population” and “The study population”consider including the categories based on nature of food and drinking facilities in the study.Consider including the systematic random sampling design and not simple random which is reflected on using by using logistic regressions rather than ordinal logistic regressions.Sample size determination and sampling techniques:How the sample size from each study area has been selected should be written more clearly as currently it is very ambiguous and unclear. Similar is true for sampling techniques and should be re-written.
**Statistical methods and Measurement of the outcome variable:**
More elaboration is required for discussing the questionnaire design” and  the“checklist” and to be included as supportive documents.The authors should state the score of each question to prevent ambiguity.Please classify the questions that were used as the basis for classification of the participants into the ones with good/poor knowledge or favorable/unfavorable attitude. Authors should consider refining how they describe the analysis - it is about the factors associated with good practices (and not both good practices and poor practices).Consider using the proportional odds model during the statistical analysisConsider writing which variables were used for the univariate or multivariate models as interest variables or for the model adjustment.Additionally, for the bivariate analysis, authors could have included variables with p-value up to 0.10 with backward or forward elimination or Other? Why P-value up to 0.1? add  reference

Data collection and quality control:

Consider adding what the observations were to be, and anticipate what the observed findings were in the results sectionConsider adding how the validity and realibility of the questionnaires were checkedPlease attach your survey tool (questionnaire using original and local language)Please also attach the raw data set 

Results:

While there are 3 tables related to infection prevention and control practicesconsider including table for the information related to knowledge/attitude questionnaire and results.Rephrase statements of the results as recommended by the reviewers"Hosmer-Lemeshow test was used to test model fitness, Multicollinearity was tested using the Variance Inflation Factor (VIF) and tolerance test." Please put their results under bracket for Hosmer-Lemeshow test, VIF and tolerance test.

Discussion

Please consider discussing studies in segregated way from global to local and put justification accordingly taking into consideration the used tools.Critically discuss reasons based on the findings. Author had highlighted methodological differences between their study vs others, but maybe need to elaborate more on the socio-cultural practices context that distinguishes the respondents from this study as compare to others, or maybe to highlight more on the local health policy and strategies to prevent COVID-19 in the study localities.The information would be more beneficial to readers to learn and improve COVID-19 prevention and control in their food service establishments. 

References:

majority of the citations in the bibliography do not have page numbers please address thatSpecifically, look at citation number 14 and 15 and address the error accordingly. 

**Minor and optional points:**

The manuscript should have line number since it makes difficult for commentThe font type should be standardized as wellImprove the quality of graphsDo you think food and drinking establishments (FDE) are facilities? If it was so, does your study uses a single facility? I think the appropriate terminology was institutions-based. Please update the throughout. You should mention the statistic numbers regarding the food handlers infected by covid?mention gold standard for food handlers from food hygiene discipline.<br />2 all studies globally or in Africa related Covid vs food handlers in background section. what kind of Food and drinking establishments that had no food handlers?Is one sampling participant represent where there were many (20-100) food handlers in the selected food and drinking establishments?Moving the operational definitions to the AnnexFigures 1, 2 and 3 seem unnecessary since they are already reported in the main text.

Please submit your revised manuscript by July 15 2021 11:59PM.  If you will need more time than this to complete your revisions, please reply to this message or contact the journal office at plosone@plos.org. Please include the following items when submitting your revised manuscript:

We look forward to receiving your revised manuscript.

Kind regards,

Lucinda Shen 

Staff Editor 

on behalf of

Ammal Mokhtar Metwally, Ph.D (MD)

Academic Editor

PLOS ONE

Journal Requirements:

3. Please provide additional details regarding participant consent. In the ethics statement in the Methods and online submission information, please ensure that you have specified:

a) whether the ethics committee approved the verbal/oral consent procedure,

b) why written consent could not be obtained, and

c) how verbal/oral consent was recorded.

If your study included minors, please state whether you obtained consent from parents or guardians in these cases. If the need for consent was waived by the ethics committee, please include this information.

Reviewers' comments:

Reviewer's Responses to Questions

**Comments to the Author**

1. Is the manuscript technically sound, and do the data support the conclusions?

Reviewer #1: Yes

Reviewer #2: Yes

Reviewer #3: Yes

Reviewer #4: Partly

Reviewer #5: Yes

Reviewer #6: Partly

Reviewer #7: Partly

Reviewer #8: Partly

Reviewer #9: Partly

Reviewer #10: Yes

Reviewer #11: Yes

2. Has the statistical analysis been performed appropriately and rigorously? 

Reviewer #1: Yes

Reviewer #2: Yes

Reviewer #3: Yes

Reviewer #4: No

Reviewer #5: Yes

Reviewer #6: Yes

Reviewer #7: Yes

Reviewer #8: I Don't Know

Reviewer #9: Yes

Reviewer #10: Yes

Reviewer #11: Yes

3. Have the authors made all data underlying the findings in their manuscript fully available?

Reviewer #1: Yes

Reviewer #2: Yes

Reviewer #3: Yes

Reviewer #4: Yes

Reviewer #5: Yes

Reviewer #6: No

Reviewer #7: No

Reviewer #8: Yes

Reviewer #9: No

Reviewer #10: Yes

Reviewer #11: Yes

4. Is the manuscript presented in an intelligible fashion and written in standard English?

Reviewer #1: Yes

Reviewer #2: Yes

Reviewer #3: Yes

Reviewer #4: Yes

Reviewer #5: Yes

Reviewer #6: Yes

Reviewer #7: Yes

Reviewer #8: No

Reviewer #9: No

Reviewer #10: Yes

Reviewer #11: Yes

5. Review Comments to the Author

Reviewer #1: Hello, here are my comments that might improve the write-up:

Page 9, you mentioned about the questionnaire and checklist used during the survey. could you please elaborate more on the domains that you measured in your instrument? When I read the results section then only I realised there were domains in knowledge, attitude and practice (KAP). How each of these domains were measured? Do each domain produced score? max score? and at the end how you measure positive attitude? I noticed you had defined favourable attitude which I need to scroll up at the operational definition but it would be better if you could explain in detail about your instrument and put all the relevant definitions in the 'instrument used' section.

In page 10, you mentioned data were analysed with binary logistic regression method, maybe you can add information about the dependance binary outcome used in the analysis? and what are the independence variables that you included in the model for crude and adjusted. The statistical method was explained sophisticatedly but reader will be enlighten with additional information about variables included in the model. I'm confused whether the dependence measured in this study was knowledge or practices? How many models were tabulated?

In general for the Discussions section, instead of comparing with other studies, I would suggest if the author could critically discuss reasons based on the findings. Author had highlighted methodological differences between their study vs others, but maybe need to elaborate more on the socio-cultural practices context that distinguishes the respondents from this study as compare to others, or maybe to highlight more on the local health policy and strategies to prevent COVID-19 in the study localities. The information would be more beneficial to readers to learn and improve COVID-19 prevention and control in their food service establishments.

The rest looks good. All the best.

Reviewer #2: Abstract

I suggest that the authors consider replacing "prevent" with "reduce" - "Thus, ensuring infection prevention and control practices for this high-risk group is urgently required to prevent COVID-19 transmission."

I suggest deleting this which is not quite necessary for an Abstract - "Variables that had a p-value < 0.25 from the bivariate analysis were retained in multivariable logistic regression analysis for controlling confounders. From the multivariable analysis, variables that had a p-value < 0.05 were taken as factors significantly associated with good infection prevention and control practices to prevent COVID-19"

Delete ".... whereas the poor practice of infection prevention and control was 56.1% (95% CI: 51.4%-60.8%)" since the first part of the sentence already captures the main result.

I suggest re-arranging the results. Report proportions for knowledge, attitude and then finally practice. Then report the results for the multivariable logistic regression on factors associated with good practice.

Although attitude was lower, it may not be related to poor practices. I think the authors might be better off suggesting how the uptake of good practices can be increased?

Methods

Could the authors clarify why the selection approach focused on "houses". i.e. "To select the house-paired with 8

one eligible study participant, the data collector started with a bench mark of the known location and then walked straight forward to identify each house." Did the authors mean "premises"?

I think there were three outcomes of interest: (i) Knowledge, (ii) Attitudes and (iii) Practices, not just one. It is just that the multivariable regression analysis was conducted on (iii) Practices? I think the authors should consider refining how they describe the analysis - it is about the factors associated with good practices (and not both good practices and poor practices).

Could the authors provide justifications to support why the cut-off for KAP scores was set at the mean of the total scores? That just simply divides participants into two almost equal groups (assuming the distribution is normal). Would it be better to the authors considered a higher cut-off, say 75% of the possible score to be considered as "good"?

I am wondering whether the authors might be open to assessing whether there is a relationship between knowledge and attitude scores, and whether high knowledge and attitude scores were associated with high practice scores?

Results

Table 3 - Can the authors fill in all the missing p-values for the stratum specific estimates?

Figures 1, 2 and 3 seem unnecessary since they are already reported in the main text.

Reviewer #3: There are some typographical errors that need to be addressed.

There are statements do not have clarity in understanding and needs to be revised.

Also, majority of the citations in the bibliography do not have page numbers please address that

Specifically, look at citation number 14 and 15 and address the error accordingly.

Some technical language errors have been raised in the attached document

Reviewer #4: Thank you for analysing this interesting dataset during the COVID-19 pandemic. However, in my point of view, some fundamental issues are missing, which will improve the statistical analysis and the result of the study and as a consequence the study conclusion.

• It is not clear why only Dessie and Kombolcha town were included in the study.

• A flowchart would add useful information to the reader of the paper. Particularly because not the entire food establishments were included.

• Dichotomising the fifteen-infection scale at an arbitrary cut-off (suing the sample mean) reduces the statistical power by the same amount as would discarding a third of the data (1, 2). In general, there is no good reason to believe that there is an underlying dichotomy, and if one exists there is no reason why it should be at the mean especially as the fifteen-infection scale is not normally distributed. The fifteen-infection scale looks like an ideal candidate for one of the ordinal cumulative probability models. Therefore, I suggest using the proportional odds model.

• The authors have ignored the systematic random sampling design by using ordinal logistic regressions. I recommend considering use survey data analysis software when performing logistic regressions. This is because it is important to take into account the differences between the study design that you have used to collect the data and simple random sampling.

• The authors have used a cut-off value of 0.25 for variable selection in the multivariate model, because they believe that it is not legitimate to include a predictor with a p-value>0.25. If the authors insist on basing the inclusion rule on a p-value<0.25, then a reasonable cut-off that does allow for deletion of some variables is 0.5 (3).

• I don’t believe that the sample size calculation is relevant as the underlying primary aim is the multivariate regression modeling.

1- On the practice of dichotomization of quantitative variables. MacCallum RC, Zhang S, Preacher KJ, Rucker DD Psychol Methods. 2002 Mar; 7(1):19-40.

2- Cohen J. The cost of dichotomization. Appl Psychol Meas 1983;7: 249-53.

3- E. W. Steyerberg, M. J. C. Eijkemans, F. E. Harrell, and J. D. F. Habbema. Prognostic modelling with logistic regression analysis: A comparison of selection and estimation methods in small data sets. Stat Med, 19:1059–1079, 2000. 69, 100, 286

Reviewer #5: Comments and questions

Manuscript number = PONE-D-21-12395

Title = COVID-19 Infection Prevention and Control Practices among Food Handlers in Food and Drinking Establishments of Dessie City and Kombolcha Towns in Northeastern Ethiopia

General comments

1. The manuscript should have line number since it makes difficult for comment

2. Do you think food and drinking establishments (FDE) are facilities? If it was so, does your study uses a single facility? I think the appropriate terminology was institutions-based. Please update the throughout.

3. The study setting should be better if it explains more about existed Food and drinking establishments and food handlers. So please write this way. “Food and drinking establishments that had no food handlers during the data collection period were excluded”. This sentence explains that your sampling unit is Food and drinking establishments but from your finding the study unit is food handlers. How you do it? Please rewrite the exclusion criteria. “When one eligible study participant was not available from the selected food and drinking establishments, another visit was made the same day. If they were again not available, another visit was performed the next day in order to minimize the non-response rate. If not available after the three visits, it was taken as non-respondent”. And “Food and drinking establishments that had no food handlers during the data collection period were excluded”. These two sentences were not compatible with each other. So better to correct the exclusion criteria

4. “Then, based on the source population (food handlers) for each kebele, sample size was proportionally allocated for all ten kebeles based on their number of size of the food and drinking establishments. Using the sampling frame, a systematic random sampling with a fixed interval of every house was used to select sampling unit within each kebeles”. Better to correct as Using the sampling frame, a systematic random sampling with a fixed interval of (K) to get the next Food and drinking establishments. One food handler was included from one Food and drinking establishments.

5. “The principal investigator and supervisors made daily spot-checks for completeness of the questionnaire”. AND “Each questionnaire and observational checklist was checked daily for completeness and any incomplete questionnaires or checklists were corrected in the same day by visiting the food and drinking establishments” Were repeated sentences and please avoid one of them to reduce redundancy which found line 196 and 200.

6. “Data were analyzed using a binary logistic regression model at 95% confidence interval and variables with p-value < 0.25 during the bivariate analysis were entered into a multivariable logistic regression analysis to see the relative effect of confounding variables and interaction of variables”.

Here how do you see the confounding variables? and how you report confounding variables? better to say control confounding variables

Regarding interaction of variables how you compute interaction? if it was an interaction what order of interaction? I think you didn’t do this so please delete it or write the result of the interaction report

interaction were also done for continuous outcomes, so please delete this section

7. The sentence on page 14 says “From the bivariate analysis age, educational status, years of experience, availability of COVID- 19 prevention guidelines, availability of COVID-19 infection prevention and control focal person, availability of specific budget for COVID-19 infection prevention and control, availability of PPE in food and drinking establishments, ever having taken COVID-19 infection prevention and control training and attitude on COVID-19 infection prevention and control were p-value <0.25 and retained into the multivariable analysis”. This variables should be listed as foot note of the multivariable analysis result table 3.

8. On page 13 line 251, Almost all 378 (94.3%) of food handlers knew that washing hands with soap, sanitizer and water would inhibit or kill corona virus. Please rewrite and edit your questionnaire also (Almost all 378 (94.3%) of food handlers knew that washing hands with soap, sanitizer and water would physically remove from hands, inhibit or kill corona virus.)

9. On line 251, washing of hands at what time of the day or at what frequency? Please correct it as washing hands frequently with soap

10. On table 2, Order servants and customers to practice social distancing and to avoid touch each other. Please correct as “ availability of visibly posted Order servants and customers to practice social distancing and to avoid touch each other”

11. Please attach your survey tool (questionnaire using original and local language)

12. Please also attach the raw data set

13. Please get support for language editor to make the paper very easy for readers.

Questions

1. Infection prevention and control: the two terms are quite different. Do you justify why you use the term control? Do you mean that infected persons are working there in the establishments?

2. The tittle was about practice but the finding section has also knowledge and attitude. Better to correct title. Also how many question items were used to measure PRACTICE, KNOWLEDGE and ATTITUDE? please put the results of each of them using table with their frequency and percentage

3. On page 6 line 121, the sentence said “the study population was food handlers among the selected food and drinking establishments”. Do you include all food handlers from the selected food and drinking establishments?

4. How you reach the study participants? The sampling procedure is not clear.

5. Regarding to the exclusion criteria it said “Food and drinking establishments that had no food handlers during the data collection period were excluded”. How is an establishment open without food handlers? Who is food handler? Do owners themselves serve as food handler? Does owners that give service does not consider as food handlers? I think there is some body acting as food handler how you exclude this guy?

6. Does any Food and drinking establishments were included into the study? or those only who are licensed ?

7. Does your proportional allocation based on the number of Food and drinking establishments or the number of food handlers?

8. How many food handlers were taken from each Food and drinking establishments and how?

9. In ethical part of the document data collectors wore mask what you did if you get food handlers who do not wear mask. Don’t you give advice and supply mask.

10. On page 18 line 373 -374, what is proxy data on Page 18. What variables make respondents to made social desirability bias. does it mean you didn’t kept confidentiality/privacy and anonymity and don’t have information sheet on the questionnaire.

11. In the table1, Job position in the food and drinking establishments (Others____) please specify others job categories

Reviewer #6: Dear authors,

Thank you allowing me to review your study. I think this a good study. However, many things need to improve for the publication. Please don't be discourage, I know this paper can be improved.

1) There a lot of grammatical error through the manuscript, so my suggestion this manuscript should be send to English editing.

2) The font type should be standardized as well.

3) You should mention the statistic numbers regarding the food handlers infected by covid maybe?

4) I already read the results. I think you should compare "N" between food handlers who had experience with Covid and also without Covid. Thus, you can get the result either their practice " good" or " poor". Furthermore, that more accurate if you want to suggest any prevention and control measure for food handlers. Yes, it's major modification. but surely you can do better.

5)The figures are nice, but it's just the descriptive. Which is the readers can get it from the table.

Reviewer #7: Present study was planned to determine the level of food handlers' knowledge, attitudes, and practices in relation to COVID-19 in northeastern Ethiopia. Food handlers can be highly exposed to the virus due to their daily contacts with customers who visit their food and drinking establishments. Therefore, the information provided in this study can guide relevant training and policy making organisations in prioritizing the protection and avoiding occupational exposure. Although this study is interesting for the reader, the method and result look ambiguous. While there are 3 tables related to infection prevention and control practices, there is little information related to knowledge/attitude questionnaire and results. It is unclear which questions are used as the basis for classification of the participants into the ones with good/poor knowledge or favorable/unfavorable attitude. Therefore, major revision is required to clarify more details in these sections.

Reviewer #8: please include

1 mention gold standard for food handlers from food hygiene discipline.

2 all studies globally or in Africa related Covid vs food handlers in background section.

3. what kind of Food and drinking establishments that had no food handlers?

4. Is one sampling participant represent where there were many (20-100) food handlers in the selected food and drinking establishments?

5. How and where you adopted this standard "Good infection prevention and control practiceswas determined for study participants who responded above or equal to the mean value amongfifteen infection prevention and control practices questions, whereas poor infection preventionand control practices refers to those study participants who responded to fifteen infection prevention and control practices questions below the mean value."

6. Was validity and realibility of the questionnaires checked?

7. how did you distinguish data collectors and supervisors during recruition? Because they have the same status.

8. how did you pre-tested and assured the data quality the 10 % of selected food handlers?

9. if one study participant answer 8 question and missed answer about wearing mask and hand washing how did you accomodate? Did he/she have good knowledge?

10 you did "Hosmer-Lemeshow test was used to test model fitness. Multicollinearitywas tested using the Variance Inflation Factor (VIF) and tolerance test." Please put thier resul under bracket for Hosmer-Lemeshow test, VIF and tolerance test.

11 please format randendency one-fifth 96 (23.9%). Put only figure and percent through out result section.

12 From 11 questions how did you manage " I do not know" response?

13 in Attitude section, did you use likert scale(strongly disagreed,agreed

14. Please polish your english and your write up through out your document it needs critical improvements. look your statements e.g. "Most respondents 389 (97%) reported that when they sneezed, they coveredtheir nose and mouth with an elbow.

& Of 36 food handlers who did notwear PPE, 10 of them thought that wearing PPE is not always necessary to prevent COVID-19."

15 In your discussion part you merged studies and the reason of dicrepancy but you have to discuss in segregated way from global to local and put reason of dicrepancy accordingly.

16 when you reason out the reason of dicrepancy you are not sure about your conclusion. For example "This may be due to the differences in the tool used for assessment of..." "may be"

Reviewer #9: Andualem et al., Review: COVID-19 Infection Prevention and Control Practices among Food Handlers of Food and Drinking Establishments of Dessie City and Kombolcha Towns in Northeastern Ethiopia

Dear Editor

The manuscript titled “COVID-19 Infection Prevention and Control Practices among Food Handlers of Food and Drinking Establishments of Dessie City and Kombolcha Towns in Northeastern Ethiopia” presented by Atsedemariam Andualem and co-authors is aimed to highlight the COVID-19 prevention and control practices adopted by food handlers in two cities of Ethiopia. The authors have contributed well by selecting the important sector of society which can play a significant role towards the transmission and spread of C COVID-19 pandemic. But while reviewing the article there are some ambiguities and questions which needs to be addressed before the article will be assessed for its publication in PLOS ONE. The details of comments and queries are given below. I hope it will prove to be helpful for the authors.

Section wise Comments:

Abstract: The first paragraph of Abstract couldn’t successfully generate the hypothesis of the study. The author should clearly write atleast 2 to 3 aims and objectives to justify that why the study has been conducted? The following line is too early to be stated in the beginning therefore should be deleted from here: {Thus, ensuring infection prevention and control practices for this high-risk group is urgently required to prevent COVID-19 transmission}.

In the second paragraph of Abstract it is important a brief description should be added about the design of the questionnaire and checklist with its strength and important parameters while the description about statistical tests should be given in one line only.

In the last paragraph of Abstract the first line {Just more than one-third of food handlers had good infection prevention and control practices} should be deleted as it seems out of context. The paragraph should be made brief as there is a lot of repetition of words/phrases “infection prevention and control”. It is suggested that only the following text should be given in the last part: {Our findings showed that three-fourths of food handlers had good knowledge and more than half had a favorable attitude about infection prevention and control. Thus, integrated work is urgently needed to prevent COVID-19 transmission by further improving food handlers’ good practice of infection prevention and control}.

Keywords: should be written in an alphabetical order.

Introduction: the first 2- paragraphs should be merged with the following text added at the end of first paragraph while the rest of the text should be deleted as it is only a repetition: {According to a World Health Organization (WHO) report on 20 September 2020, 30.6 million positive cases with 950,000 deaths worldwide have been recorded. On that date, Africa accounted for 1,145,397 COVID-19 positives with a total death of 24,757; and in Ethiopia the number of COVID-19 cases reached 68,131 and confirmed deaths reached 1,089 [9].}

In the last paragraph the following sentence should be either deleted or re-written as it does not clearly interpret what is author trying to convey: {Implementation of IPC practices based on WHO and Ethiopian Ministry of Health (EMoH) guidelines is rapid, decisive, and collective action which saves millions of lives [13, 16, 17].}

The objectives of the study should be elaborated more by focusing on what the authors are trying to achieve by conducting this study and how it will be helpful for the society as the given objective: {Therefore, this study was designed to assess the practice of infection prevention and control strategies against the novel coronavirus pandemic among food handlers in Dessie City and Kombolcha Town food and drinking establishments} is too generalized.

Methods and Materials:

Study Area: Add some description about the socioeconomic status of the area to generate some idea about the general people perceptions and attitudes in the study area.

Study design, period, and population:

The terms “The source population” and “The study population” leads to confusion as they both refer to study subjects therefore it would be better to use “The study population” to avoid confusion. The following lines should be re-written as seems a repetition {The source population was all food handlers working in Dessie City and Kombolcha Town food and drinking establishments. The study population was food handlers among the selected food and drinking establishments}.

It is also important that detail data in the form of table/graph (could be added in Annexure) should be given about the categories based on nature of food and drinking facilities included in the study.

Sample size determination and sampling techniques:

How the sample size from each study area has been selected should be written more clearly as currently it is very ambiguous and unclear. Similar is true for sampling techniques and should be re-written.

Measurement of the outcome variable:

It is important to discuss questionnaire and important aspects included in its structure and how the mean value has been calculated. As currently the most important tool of the study “Questionnaire design” and “checklist” has been lacking from the study which is a serious discrepancy.

Operational definitions:

These definitions should be given in the Annexure.

Data collection and quality control:

This part could also be given in the Annexure.

Results:

Overall, the results have been interpreted in detail but there is a need to improve the quality of graphs and tables.

Discussion:

At large Discussion is well written and has thoroughly justified the results.

Overall, some grammatical mistakes have been found in the write up along with repetition of words and phrases therefore it is highly recommended that authors should proof read the draft from an expert with strong English language background to improve the overall quality of the manuscript.

Reviewer #10: The article of Metadel et al. is a completed study and is of interest to scientists for studying the problems of COVID-19 iinfection prevention and control practices among food handlers. The manuscript describes the the knowledge, practice and attitudes of handlers of food and drinking establishments’ in cities and towns of Ethiopia The paper is well structured but it needs some modification prior to publication. I think this manuscript should be accepted for publication after some revision:

1.It is not clearly written in methods, which variables were used for the univariate or multivariate models as interest variables or for the model adjustment.

2.Additionally, for the bivariate analysis, authors could have included variables with p-value up to 0.10 with backward or forward elimination or Other? Why P-value up to 0.1? Do you have reference?

3.Authors must provide more details on methods: which factors and explanatory variables were addressed by the questionnaire and in the analyses (e.g., as continuous or dichotomous).

4. The manuscript has used inappropriate English word “illiterate” which is erroneous that must be removed.

5. It was a kind of cross sectional study, why you used prolonged time (two months) for data collections?

6.Why the experiences were defined as <1, >1-5 and >5 years? It should be clearly stated in methods.

Reviewer #11: There are some observations made in the process of this review.

1. Methods

The authors mentioned in the section of measures of measures that the mean value from fifteen questions was considered good infection prevention and control practice. The authors should state the score of each question to prevent ambiguity. Readers will be well informed of the for instance, the authors says either each question contains 1,3,3 or so score, from which the mean value I'd calculated.

In the data collection and quality control section, authors stated that, protested questions and on the spot observations checklist were used. However, they failed readers of what was entailed on the observations checklist. Readers will appreciate it much, if they are informed what the observations were to be, and anticipate what the observed findings were in the results section

2. The results

The authors did not indicate the observations made by way of the observations checklist that was employed. instead the results were focused on only the structure question. It is important for readers to know whether answers provide by the study subjects were in concordance with the observations made by the researchers. This will probably prevent some bias introduced by the subjects because of the researchers' presence.

In reporting the the results, the third sentence should be checked and rephrase...." most of the respondents were under 30 years 273 (68.1%)."

"The odds of having good COVID-19 infection prevention and control practice among food handlers who had work experience of greater than 5 years were 3 times (AOR=2.55; CI:1.43-5.77) greater compared to those who had 5 or fewer years of work experience. " this statement should be written to reflect the results. It is emphatic, however, that is not the case. Trying saying "almost 3 times....."

3. Discussion

In the second paragraph, the researchers identified some difference in results and attributed same to differences in tools

"This may be due to the differences in the tool used for assessment of COVID-19 infection prevention and control practice, differences in study participants where our study focused on food handlers, or a methodological difference such as the difference in the accessibility of COVID-19 infection prevention and control guidelines and types of PPE in the food and drinking establishments. "

It is therefore important to mention the tool used in your study and that used in the other studies which could be the plausible reason.

6. PLOS authors have the option to publish the peer review history of their article (what does this mean?). If published, this will include your full peer review and any attached files.

Reviewer #1: **Yes: **Mohd Hatta Bin Abdul Mutalip

Reviewer #2: No

Reviewer #3: No

Reviewer #4: No

Reviewer #5: No

Reviewer #6: No

Reviewer #7: No

Reviewer #8: **Yes: **Thomas Ayalew Abebe

Reviewer #9: **Yes: **Dr. Zaira Ahmad

Reviewer #10: **Yes: **Tariku Neme Afata, Jimma University, department of environmental health science and technology

Reviewer #11: No

---

## [Author Response · Author response to Decision Letter 0]

19 Aug 2021

Rebuttal letter

Response to reviewers for the paper PONE-D-21-12395

COVID-19 Infection Prevention Practices among Food Handlers in Food and Drink Establishments of Dessie City and Kombolcha Town, Northeastern Ethiopia 

Dear Academic Editor. Thank you for your summarized main concerns and suggestions that need to be addressed, and we updated the manuscript accordingly. 

Here are the detail answers for your concerns: 

Question #1. Overall General comment:

Please note that further language improvements is indicated. Consider revising the spelling, grammar, diction, and syntax throughout the manuscript for increased clarity. The paragraphs should be made brief as there is a lot of repetition of words/phrases (refer to reviewer

specific comments for some sentences as example). Consider replacing some words as described by the reviewers to be more specific and precise e.g : in the Abstract consider replacing "prevent" with "reduce" - "Thus, ensuring infection prevention and control practices

for this high-risk group is urgently required to prevent COVID-19 transmission."

Response: Thank you for the critical observation, we improved the language of the manuscript based on the comments given.

Main points to be addressed:

Question #2. Abstract: 

Consider adding one paragraph as background discussing the hypothesis of the study.

Formulate the objective with 2 to 3 aims Delete the non-related conclusion¨ Thus, ensuring infection prevention and control practices for this high-risk group is urgently required to prevent COVID-19 transmission}. Consider adding the study design and too to the methodology and remove the statistical tests description. Consider re-arranging the results Report proportions for knowledge, attitude and then finally practice. Then report the results for the multivariable logistic regression on factors associated with good practice. Although attitude was lower, it may not be related to poor practices. I think the authors might be better off suggesting how the uptake of good practices can be increased? Consider removing from the result of the abstract sentences that out of context. Delete ".... whereas the poor practice of infection prevention and control was 56.1% (95% CI: 51.4%-60.8%)" since the first part of the sentence already captures the main result. restrict to provide conclusions out of the study ¨Our findings showed that three-fourths of food handlers had good knowledge and more than

half had a favorable attitude about infection prevention and control. Thus, integrated work is urgently needed to prevent COVID-19 transmission by further improving food handlers’ good practice of infection prevention and control}. 

Response: Thank you for the comments. The hypothesis of the study is modified, see line 21 and 22 and we added objective of the study, see line 22-25. We modified the method part of the abstract as your suggestion; see from line 27-34. From the result part, our study main concern is infection prevention (IP) practices not KAP; knowledge and attitude are as factors that affect IP practices leads to put percentage of good IP primarily and the poor practice percentage is deleted. Although food handlers have favorable attitude, they may not practice IP strategies properly due to different factors like shortage of time, unsafe environment…but after a while if the constraints are reduced favorable attitude dramatically increases good IP practices. We modified the conclusion part of abstract, see from line 45-49.

Question #3. Consider writing the keywords in an alphabetical order.

Response: We arranged them alphabetically

Question #4. Introduction:

 The first 2- paragraphs should be merged with the following text added at the end of first paragraph while the rest of the text should be deleted as it is only a repetition: {According to a World Health Organization (WHO) report on 20 September 2020, 30.6 million positive cases with 950,000 deaths worldwide have been recorded. On that date, Africa accounted for 1,145,397 COVID-19 positives with a total death of 24,757; and in Ethiopia the number of COVID-19 cases reached 68,131 and confirmed deaths reached 1,089 [9].

In the last paragraph the following sentence should be either deleted or re-written as it does not clearly interpret what is author trying to convey: {Implementation of IPC practices based on WHO and Ethiopian Ministry of Health (EMoH) guidelines is rapid, decisive, and

collective action which saves millions of lives [13, 16, 17].}

Response: Thank you. We corrected the two paragraphs as you said and {Implementation of IPC practices based on WHO and Ethiopian Ministry of Health (EMoH) guidelines is rapid, decisive, and collective action which saves millions of lives [13, 16, 17].} is deleted.

Question #5. Objectives

Consider formulating the objectives of the study to be more focused as the given objective is too generalized.

Response: Thank you for the pertinent comment. We updated the objective of the study as per your suggestion please see the revised version in the abstract page 2, line number 22-25 and from Introduction page 5 line 91 to 94.

Methodology

Question #6. Study Area:

It is not clear why only Dessie and Kombolcha town were included in the study. Add some description about the socioeconomic status of the area to generate some idea about the general people perceptions and attitudes in the study area.

Response: 

- As we justify the study from introduction part page 5 line 87 to 91; Dessie City and Kombolcha Town are the main hotspot areas for COVID-19 due to the nearby road from Djibouti (a neighbouring country with high prevalence of COVID-19). Both residents of the towns and travellers use food and drink establishments together in Dessie City and Kombolcha Town, increasing the transmission of COVID-19.

- We included the socioeconomic status of the area, please see page 6; line number 104-110.

Question #7. Approach

•A flowchart would add useful information to the reader of the paper. Particularly because not the entire food establishments were included. 

Response: We respected your comment, but why we do not want to add the flow chart of sampling procedure to reduce the redundancy with the narration from Sample size determination and sampling techniques. The current version is clear and easily understandable, please see page 7 and 8; line number 117- 148.

Question #8. Study design, period, and population: consider more clarification for the terminology : “The source population” and “The study population” consider including the categories based on nature of food and drinking facilities in the study. Consider including the systematic random sampling design and not simple random which is reflected on using by using logistic regressions rather than ordinal logistic regressions.

Response: Great thanks;

- “The source population was all food handlers working in Dessie City and Kombolcha Town food and drinking establishments” is deleted to make it clear.

- We modified as you suggested.

 Question #9. Sample size determination and sampling techniques: How the sample size from each study area has been selected should be written more clearly as currently it is very ambiguous and unclear. Similar is true for sampling techniques and should be re-written.

Response: Sorry for the confusion we did. Now we make it clear and easily understandable, please see page 7 and 8, and line number 117- 148.

Question #10. Statistical methods and Measurement of the outcome variable: More elaboration is required for discussing the questionnaire design” and the“ checklist” and to be included as supportive documents. The authors should state the score of each question to prevent ambiguity. Please classify the questions that were used as the basis for classification of the participants into the ones with good/poor knowledge or favorable/unfavorable attitude. Authors should consider refining how they describe the analysis - it is about the factors

associated with good practices (and not both good practices and poor practices). Consider using the proportional odds model during the statistical analysis Consider writing which variables were used for the univariate or multivariate models as interest variables or for the model adjustment. Additionally, for the bivariate analysis, authors could have included

variables with p-value up to 0.10 with backward or forward elimination or Other? Why P value up to 0.1? add reference

Response: 

Thank you for these invaluable comments. We correct the comments as you suggested, please see page 8 line 150-156 and page 10; line number 180-188.

Typical stopping rules for explanatory modeling are p-value thresholds of 0.05 and 0.10. If a p-value is greater than the threshold, the term is removed from the model with backward LR (for our study). Our reference is available online at www.jmp.com/en_us/statistics-knowledge-portal/what-is-multiple-regression/variable-selection.

Question #11. Data collection and quality control: Consider adding what the observations were to be, and anticipate what the observed findings were in the results section Consider adding how the validity and realibility of the questionnaires were checked Please attach your survey tool (questionnaire using original and local language) Please also attach the raw data set.

Response: Thank you the comments

- From data collection and quality control: we added what the observations to be; please see page 10 line number 182 and from result section: we put ‘observation’ in bracket to differentiate observation checklist from interview at Table 2

- The internal reliability was checked using the pretest result and Content validity was done for the instrument by one senior environmental health professional and one nursing educator. Please see page 11,line number 204-209.

- The raw data set is attached as S1,S2 and S3.

Question #12. Results:

While there are 3 tables related to infection prevention and control practices consider including table for the information related to knowledge/attitude questionnaire and results. Rephrase statements of the results as recommended by the reviewers "Hosmer-Lemeshow test was used to test model fitness, Multicollinearity was tested using the Variance Inflation Factor (VIF) and tolerance test." Please put their results under bracket for Hosmer-Lemeshow test, VIF and tolerance test.

Response:

- Thank you, we add Table 4 and 5 related to knowledge and attitude questionnaire with frequency and percentage respectively.

- And we put the value in bracket. Please see line number 218-222.

Question #13. Discussion

Please consider discussing studies in segregated way from global to local and put justification accordingly taking into consideration the used tools. Critically discuss reasons based on the findings. Author had highlighted methodological differences between their study vs others,

but maybe need to elaborate more on the socio-cultural practices context that distinguishes the respondents from this study as compare to others, or maybe to highlight more on the local health policy and strategies to prevent COVID-19 in the study localities. The information would be more beneficial to readers to learn and improve COVID-19 prevention and control in their food service establishments.

Response: Thank you for these important comments, and we modified the discussion section as you suggested, please see the revised version page 16; line number 311-321.

Question #14. References: majority of the citations in the bibliography do not have page numbers please address that Specifically, look at citation number 14 and 15 and address the error accordingly.

Response: 

- Thank you, and we tried to find page numbers and updated them but some of the references have no page numbers.

- All references are updated using PLOS ONE format.

Minor and optional points:

The manuscript should have line number since it makes difficult for comment. The font type should be standardized as well Improve the quality of graphs Do you think food and drinking establishments (FDE) are facilities? If it was so, does your study uses a single facility? I think the appropriate terminology was institutions-based. Please update the throughout. You should mention the statistic numbers regarding the food handlers infected by covid? mention gold standard for food handlers from food hygiene discipline.

2 all studies globally or in Africa related Covid vs food handlers in background section. what kind of Food and drinking establishments that had no food handlers? Is one sampling participant represent where there were many (20-100) food handlers in the selected food and drinking establishments?

Moving the operational definitions to the Annex Figures 1, 2 and 3 seem unnecessary since they are already reported in the main text.

Response: Thank you for these essential questions. We made a correction for all the mentioned issues

Dear reviewers. Thank you for your valuable comments and we updated the manuscript using your comments. 

Here are the detail answers for your concerns: 

Reviewer #1: Hello, here are my comments that might improve the write-up: 

Question #1. Page 9, you mentioned about the questionnaire and checklist used during the survey. could you please elaborate more on the domains that you measured in your instrument? When I read the results section then only I realised there were domains in knowledge, attitude and practice (KAP). How each of these domains were measured? Do each domain produced score? max score? and at the end how you measure positive attitude? I noticed you had defined favorable attitude which I need to scroll up at the operational definition but it would be better if

you could explain in detail about your instrument and put all the relevant definitions in the 'instrument used' section.

Response: Thank you for these invaluable comments. We made correction as you suggested, please see page 8 line 150-156 and page 10; line 180-189.

Question #2. In page 10, you mentioned data were analysed with binary logistic regression method, may be you can add information about the dependence binary outcome used in the analysis? and what are the independence variables that you included in the model for crude and adjusted. The statistical method was explained sophisticatedly but reader will be enlighten with additional information about variables included in the model. I'm confused whether the dependence measured in this study was knowledge or practices? How many models were tabulated?

Response: 

- Sorry for the confusion. The dependence binary outcome variable is explained at page 8 line 146-151 and the description of independent variables included in the model for crude is found page 9, line number 175-177. The independent variables those included in the model for adjusted are explained at page 14, from line number 274-279. The dependent variable is Infection prevention practice with variable of interest ‘Good infection prevention practice’.

- The tabulated models were; bivariate and multivariate logistic regression model.

Question #3. In general for the Discussions section, instead of comparing with other studies, I would suggest if the author could critically discuss reasons based on the findings. Author had highlighted methodological differences between their study vs others, but maybe need to elaborate more on the socio-cultural practices context that distinguishes the respondents from this study as compare to others, or maybe to highlight more on the local health policy and strategies to prevent COVID-19 in the study localities. The information would be more beneficial to readers to learn and improve COVID-19 prevention and control in their food service establishments.

The rest looks good. All the best.

Response: Thank you dear reviewer for the comments, and we modified the discussion section as you suggested, please see the revised version page 16; line number 311-321.

Reviewer #2: 

Question #1. Abstract

I suggest that the authors consider replacing "prevent" with "reduce" - "Thus, ensuring infection prevention and control practices for this high-risk group is urgently required to prevent COVID-19 transmission." I suggest deleting this which is not quite necessary for an Abstract -

"Variables that had a p-value < 0.25 from the bivariate analysis were retained in multivariable logistic regression analysis for controlling confounders. From the multivariable analysis, variables that had a p-value < 0.05 were taken as factors significantly associated with

good infection prevention and control practices to prevent COVID-19"

 Delete ".... whereas the poor practice of infection prevention and control was 56.1% (95% CI: 51.4%-60.8%)" since the first part of the sentence already captures the main result.

I suggest re-arranging the results. Report proportions for knowledge, attitude and then finally practice. Then report the results for the multivariable logistic regression on factors associated with good practice.

Although attitude was lower, it may not be related to poor practices. I think the authors might be better off suggesting how the uptake of good practices can be increased?

Response: We replace "prevent" with "reduce" ; please see page 2 line number 23 and modified the method part of the abstract as your suggestion; see page 2 from line 27-35. From the result part, our study main concern is infection prevention (IP) practices not KAP; knowledge and attitude are as factors that affect IP practices leads to put percentage of good IP practices (our variable of interest) primarily and the poor practice percentage is deleted. Although food handlers have favorable attitude, they may not practice IP strategies properly due to different factors like shortage of time, unsafe environment…but after a while if the constraints are reduced favorable attitude dramatically increases good IP practices. 

Question #2. Methods

Could the authors clarify why the selection approach focused on "houses". i.e. "To select the house-paired with 8 one eligible study participant, the data collector started with a bench mark of the known location and then walked straight forward to identify each house." Did the authors mean "premises"?

I think there were three outcomes of interest: (i) Knowledge, (ii) Attitudes and (iii) Practices, not just one. It is just that the multivariable regression analysis was conducted on (iii) Practices? I

think the authors should consider refining how they describe the analysis - it is about the factors associated with good practices (and not both good practices and poor practices).

Could the authors provide justifications to support why the cut-off for KAP scores was set at the mean of the total scores? That just simply divides participants into two almost equal groups (assuming the distribution is normal). Would it be better to the authors considered

a higher cut-off, say 75% of the possible score to be considered as "good"?

I am wondering whether the authors might be open to assessing whether there is a relationship between knowledge and attitude scores, and whether high knowledge and attitude scores were as sociated with high practice scores?

Response: Thank you for the comments you have given.

- Sorry for the confusion, we changed “houses” to “food and drinking establishments” and it does not mean "premises". Please see page 7, line number 135-137.

- No, we have only one outcome variable that is Infection prevention (IP) practice with variable of interest good IP practice. Knowledge and attitude were factors those may associated with IP practice.

- Why we prefer mean as a cut point is that 

1. Mean is typically the best measure of central tendency because it takes all values into account (the distribution of the data is normal). 

2. We do not have references to determine the cut point other than mean that pushes us to use it as a cut of point.

3. The questions we used to measure IP practice and knowledge are yes/no item , for attitude are likert scale which cannot be exposed to bias, so the most preferable classification cut point for such items is mean.

- Really we assess the association between Knowledge and Attitude but no significant association between them. From the bivariate analysis, favorable attitude was associated with good IP practice but not Knowledge. After controlling confounders using multivariable analysis, it did not associate.

Question #3. Results

Table 3 - Can the authors fill in all the missing p-values for the stratum specific estimates?

Figures 1, 2 and 3 seem unnecessary since they are already reported in the main text.

Response: 

Ok thank you dear reviewer, we fill all the missing p-values; please see the current updated manuscript Table 7, line number 608-611. 

Really right you are, so we remove Figures 1, 2 and 3 since they are unnecessary as they are already reported in the main text.

Reviewer #3: 

Question #1. There are some typographical errors that need to be addressed. There are statements do not have clarity in understanding and needs to be revised. Also, majority of the citations in the bibliography do not have page numbers please address that Specifically, look at citation number 14 and 15 and address the error accordingly. Some technical language errors have been raised in the attached document

Response: Thank you dear reviewer, all language errors and problems in references are well corrected and addressed.

Reviewer #4: 

Thank you for analysing this interesting dataset during the COVID-19 pandemic. However, in my point of view, some fundamental issues are missing, which will improve the statistical analysis and the result of the study and as a consequence the study conclusion.

Question #1• It is not clear why only Dessie and Kombolcha town were included in the study.

Response: As we justify the study from introduction part page 5, line number 89 to 95; Dessie City and Kombolcha Town are the main hotspot areas for COVID-19 due to the nearby road from Djibouti ( a neighbouring country with high prevalence of COVID-19). Both residents of the towns and travellers use food and drink establishments together in Dessie City and Kombolcha Town, increasing the transmission of COVID-19, that is why we prefer them as the study site. 

Question #2• A flow chart would add useful information to the reader of the paper. Particularly because not the entire food establishments were included.

Response: We respected your comment, but why we do not want to add the flow chart of sampling procedure is to reduce the redundancy with the narration from Sample size determination and sampling techniques. As to us now it is clear and easily understandable, please see page 7 and 8; line number 117- 148.

Question #3• Dichotomising the fifteen-infection scale at an arbitrary cut-off (suing the sample mean) reduces the statistical power by the same amount as would discarding a third of the data (1, 2). In general, there is no good reason to believe that there is an underlying dichotomy, and if one exists there is no reason why it should be at the mean especially as the fifteen-infection scale is not normally distributed. The fifteen-infection scale looks like an ideal candidate for one of the ordinal cumulative probability models. Therefore, I suggest using the proportional odds model.

Response: There were no references to adopt cut off point other than central tendency measurement since COVID-19 was new and emergent issue but not well studied.

1. Mean is typically the best measure of central tendency because it takes all values into account and the distribution of our data was normal. 

2. The questions we used to measure IP practice were yes/no item , no was denoted by “0” and yes was denoted by “1” which cannot be exposed to bias, so the most preferable classification cut point for such items is mean.

Question #4• The authors have ignored the systematic random sampling design by using ordinal logistic regressions. I recommend considering use survey data analysis software when performing logistic regressions. This is because it is important to take into account the differences between the study design that you have used to collect the data and simple random sampling.

Response: We did not used ordinal logistic regression. Our model is binary logistic regression. Please see the data analysis section. 

Question #5. The authors have used a cut-off value of 0.25 for variable selection in the multivariate model, because they believe that it is not legitimate to include a predictor with a p-value>0.25. If the authors insist on basing the inclusion rule on a p-value<0.25, then a

reasonable cut-off that does allow for deletion of some variables is 0.5 (3).

Response: Thank you for this key comment. When we include all variables that has a p-value of <0.3, the model was not fit compared to the models that has included variables p-value <0.25. That is why we used a p-value <0.25

 Question #6. I don’t believe that the sample size calculation is relevant as the underlying primary aim is the multivariate regression modeling.

Response: The sample size is estimated based on scientific procedure with the assumptions of sample size determinations. We believe that our sample size sufficient despite it is always good to have a large ample size. 

Reviewer #5: Comments and questions

Question #1. The manuscript should have line number since it makes difficult for comment

Response: Thank you for your concern. We insert line numbers

Question #2. Do you think food and drinking establishments (FDE) are facilities? If it was so, does your study uses a single facility? I think the appropriate terminology was institutions-based. Please update the throughout. 

Response: Thank you. We take your advice since you are correct that institution is an established organization, especially one dedicated to education, public service, culture or the care of the destitute, poor etc..

Question #3. The study setting should be better if it explains more about existed Food and drinking establishments and food handlers. So please write this way. “Food and drinking establishments that had no food handlers during the data collection period were excluded”. This sentence explains that your sampling unit is Food and drinking establishments but from your finding the study unit is food handlers. How you do it? Please rewrite the exclusion criteria. “When one eligible study participant was not available from the selected food and drinking establishments, another visit was made the same day. If they were again not available, another visit was performed the next day in order to minimize the non-response rate. If not available after the three visits, it was taken as non-respondent”. And“ Food and drinking establishments that had no food handlers during the data collection period were excluded”. These two sentences were not compatible with each other. So better to correct the exclusion criteria

Response: Yes, our sampling unit is either of Food and drinking establishments or food handlers since one food handler was selected from each of the selected food and drinking establishment. 

Thank you, the exclusion criteria is to mean “Food and drinking establishments that had no eligible food handlers during the data collection period after three visits were excluded”. 

Question #4. “Then, based on the source population (food handlers) for each kebele, sample size was proportionally allocated for all ten kebeles based on their number of size of the food and drinking establishments. Using the sampling frame, a systematic random sampling with a fixed interval of every house was used to select sampling unit within each kebeles”.Better to correct as Using the sampling frame, a systematic random sampling with a fixed interval of (K) to get the next Food and drinking establishments. One food handler was included from one Food and drinking establishments.

Response: We correct it as “Using the sampling frame, a systematic random sampling with a fixed interval of (K) was used to get the next Food and drinking establishments within each kebele”.

Question #5. “The principal investigator and supervisors made daily spot-checks for completeness of the questionnaire”. AND “Each questionnaire and observational checklist was checked daily for completeness and any incomplete questionnaires or checklists were corrected in the same day by visiting the food and drinking establishments” Were repeated sentences and please avoid one of them to reduce redundancy which found line 196 and 200.

Response: Sorry for the redundancy. We omit the sentence that stats “The principal investigator and supervisors made daily spot-checks for completeness of the questionnaire”.

Question #6. “Data were analyzed using a binary logistic regression model at 95% confidence interval and variables with p-value < 0.25 during the bivariate analysis were entered into a multivariable logistic regression analysis to see the relative effect of confounding variables and interaction of variables”.

 Here how do you see the confounding variables? and how you report confounding variables? better to say control confounding variables 

 Regarding interaction of variables how you compute interaction? if it was an interaction what order of interaction? I think you didn’t do this so please delete it or write the result of the interaction report 

 interaction were also done for continuous outcomes, so please delete this section 

Response: Thank you. We correct it as “Data were analyzed using a binary logistic regression model at 95% confidence interval and variables with p-value< 0.25 during the bivariate analysis were entered into a multivariable logistic regression analysis to control confounding variables”.

Question #7. . On page 13 line 251, Almost all 378 (94.3%) of food handlers knew that washing hands with soap, sanitizer and water would inhibit or kill corona virus. Please rewrite and edit your questionnaire also (Almost all 378 (94.3%) of food handlers knew that washing hands with soap, sanitizer and water would physically remove from hands, inhibit or kill corona virus.) 

Response: Thank you. We correct it as “Around 378 (94.3%) of food handlers knew that washing hands frequently with soap, sanitizer and water would physically remove, inhibit or kill coronavirus from hands”.

Question #8. On line 251, washing of hands at what time of the day or at what frequency? Please correct it as washing hands frequently with soap

Response: Thank you. We correct it

Question #9. . On table 2, Order servants and customers to practice social distancing and to avoid touch each other. Please correct as “availability of visibly posted Order servants and customers to practice social distancing and to avoid touch each other”

Response: Thank you. We correct it as you suggested “Availability of visibly posted order for servants and customers to practice social distancing and to avoid touch each other”.

Question #10. . Please attach your survey tool (questionnaire using original and local language)

 Response: We attached it

Question #11. . Please also attach the raw data set 

 Response: We attached it

Question #12. . Please get support for language editor to make the paper very easy for readers.

Response: We got language editing service from the first English language speaker and Lisa Penttila. We acknowledged the editor in the acknowledgement section as follows. The last but not the least we also thank the first English language speaker and Lisa Penttila for language editing of the manuscript.

Questions 

1. Infection prevention and control: the two terms are quite different. Do you justify why you use the term control? Do you mean that infected persons are working there in the establishments? 

Response: We corrected it as Infection Prevention Practices …

2. The tittle was about practice but the finding section has also knowledge and attitude. Better to correct title. Also how many question items were used to measure PRACTICE, KNOWLEDGE and ATTITUDE? please put the results of each of them using table with their frequency and percentage

Response: Thank you for your critical observation.

We used knowledge and attitude as factors affecting IPC practice, so we did not include knowledge and attitude from the title. 

The number of question items to measure KNOWLEDGE and ATTITUDE are found from operational definition and PRACTICE from measurement of the outcome variable as 

1. Good knowledge refers to those study participants who correctly answered more than or equal to the mean score of eleven knowledge questions. 

2. Favorable attitude refers to those study participants who scored greater than or equal to the mean score of nine attitude questions.

3. Good infection prevention practices was determined for study participants who responded above or equal to the mean value among fifteen infection prevention practices questions.

And the frequency and percentage of each practice, knowledge and attitude question is put using tables.

3. On page 6 line 121, the sentence said “the study population was food handlers among the selected food and drinking establishments”. Do you include all food handlers from the selected food and drinking establishments?

Response: No, only one food handler was included from one Food and drinking establishments.

4. How you reach the study participants? The sampling procedure is not clear.

Response: The numbers of food and drinking establishments from Dessie city and Kombolcha Town were proportionally allocated based on their number of food and drinking establishments. Using the sampling frame, a systematic random sampling with a fixed interval of (K) was used to get the next Food and drinking establishments within each town. To select the food and drinking establishment-paired with one eligible study participant, the data collector started with a bench mark of the known location and then walked straight forward to identify each food and drinking establishment.

Furthermore, when more than one eligible study participant was present in the selected food and drinking establishments, a lottery method was used to select one study participant to estimate the proportion of good or poor practice of infection prevention in the study area relative to the total sample size. When one eligible study participant was not available from the selected food and drinking establishments, another visit was made the same day. If they were again not available, another visit was performed the next day in order to minimize the non-response rate. If not available after the three visits, it was taken as non-respondent.

5. Regarding to the exclusion criteria it said “Food and drinking establishments that had no food handlers during the data collection period were excluded”. How is an establishment open without food handlers? Who is food handler? Do owners themselves serve as food handler?Does owners that give service does not consider as food handlers? I think there is some body acting as food handler how you exclude this guy?

Response: Really you are right, we correct it as Food and drinking establishments that had no eligible food handlers during the data collection period were excluded. Eg Data collectors got food handlers who are less than 18 years old in Food and drinking establishments during data collection period were excluded after three visits.

6. Does any Food and drinking establishments were included into the study? or those only who are licensed ? 

Response: Those licensed Food and drinking establishments were included in our study.

7. Does your proportional allocation based on the number of Food and drinking establishments or the number of food handlers?

Response: In our study, proportional allocation of sample size was based on the number of food and drinking establishments in each town not the number of food handlers.

8. How many food handlers were taken from each Food and drinking establishments and how?

Response: Only one food handler was selected from each Food and drinking establishments. To select the food and drinking establishment-paired with one eligible study participant, the data collector started with a bench mark of the known location and then walked straight forward to identify each house. Furthermore, when more than one eligible study participant was present in the selected food and drinking establishments, a lottery method was used to select one study participant to estimate the proportion of good or poor practice of infection prevention in the study area relative to the total sample size. When one eligible study participant was not available from the selected food and drinking establishments, another visit was made the same day. If they were again not available, another visit was performed the next day in order to minimize the non-response rate. If not available after the three visits, it was taken as non-respondent.

9. In ethical part of the document data collectors wore mask what you did if you get food handlers who do not wear mask. Don’t you give advice and supply mask.

Response: We gave health education about the importance of mask, sanitizer… (PPE) and if there were no any access of PPE, we gave mask, sanitizer, soap……to them.

10. In the table1, Job position in the food and drinking establishments (Others____) please specify others job categories 

Response: Others: Butchers, Kitchen manager and Vegetable station workers

Reviewer #6: Dear authors, 

Thank you allowing me to review your study. I think this a good study. However, many things need to improve for the publication. Please don't be discourage, I know this paper can be improved.

Question #1. There a lot of grammatical error through the manuscript, so my suggestion this manuscript should be send to English editing.

Response: We got language editing service from the first English language speaker and Lisa Penttila. Therefore the whole documents are modified.

Question #2. The font type should be standardized as well.

Response: Thank you dear reviewer, we updated it well.

Question #3. You should mention the statistic numbers regarding the foodhandlers infected by covid maybe?

Response: Fortunately we did not get food handlers infected by COVID-19

Question #4. I already read the results. I think you should compare "N" between food handlers who had experience with Covid and also without Covid. Thus, you can get the result either their practice " good" or " poor". Furthermore, that more accurate if you want to suggest any prevention and control measure for food handlers. Yes, it's major modification. but surely you can do better.

Response: Thank you, and fortunately there were no food handlers with COVID-19, so we compare food handlers with good IP practices with that of poor practice

Question #5. The figures are nice, but it's just the descriptive. Which is the readers can get it from the table.

Response: Thank you; figures are already omitted since it is described within the document.

Reviewer #7: 

Present study was planned to determine the level of food handlers' knowledge, attitudes, and practices in relation to COVID-19 in northeastern Ethiopia. Food handlers can be highly exposed to the virus due to their daily contacts with customers who visit their food and drinking establishments. Therefore, the information provided in this study can guide relevant training and policy making organizations in prioritizing the protection and avoiding occupational exposure.

Question #1. Although this study is interesting for the reader, the method and result look ambiguous. While there are 3 tables related to infection prevention and control practices, there is little information related to knowledge/attitude questionnaire and results. It is unclear which questions are used as the basis for classification of the participant into the ones with good/poor knowledge or favorable/unfavorable attitude. Therefore, major revision is required to clarify more details in these sections.

Response: We appreciate your concern. The method is modified to make it clear; please see from line number 99-212 and we add Table 4 and 5 related to knowledge/attitude, please see from line number 560-576.

Reviewer #8: please include

Question #1. all studies globally or in Africa related Covid vs food handlers in background section.

Response: Thank you, and please see the revised version

Question #2. what kind of Food and drinking establishments that had no food handlers?

Response: Sorry for the confusion we made, no food and drinking establishments without food handler but we want to say Food and drinking establishments that had no eligible food handlers (food handlers whose age 18 years and above) during the data collection period were excluded. Please see page 6; line number 111-112.

Question #3. Is one sampling participant represent where there were many (20-100) food handlers in the selected food and drinking establishments?

Response: Right you are, but when we performed preliminary assessment before conducting this research, there were no food and drinking establishments that has more than 10 food handlers (most of them has 2-6). That is why one eligible food handler was selected from each food and drinking establishment by lottery method.

Question #4. How and where you adopted this standard "Good infection prevention and control practiceswas determined for study participants who responded above or equal to the mean value among fifteen infection prevention and control practices questions, whereas poor infection prevention and control practices refers to those study participants who responded to fifteen infection prevention and control practices questions below the mean value."

Response: There were no references to adopt cut off point other than central tendency measurement since COVID-19 was new and emergent issue but not well studied.

3. Mean is typically the best measure of central tendency because it takes all values into account and the distribution of our data was normal. 

4. The questions we used to measure IP practice and knowledge are yes/no item , for attitude are likert scale which cannot be exposed to bias, so the most preferable classification cut point for such items is mean.

Question #5. Was validity and realibility of the questionnaires checked?

Response: Yes we checked, please see the modified document page 11; line number 198-201.

Question #6. how did you distinguish data collectors and supervisors during recruition? Because they have the same status.

Response: Even though both data collectors and supervisors had a status of Bsc degree, data collectors were Bsc nurse who had previous experience of COVID-19 data collection and supervisors had Bsc in Environmental Health with previous experience of COVID-19 data collectors’ supervision activities. Please see page 10; line number 185-187.

Question #7. how did you pre-tested and assured the data quality the 10 % of selected food handlers?

Response: We thank for this important question, and we pretested the questionnaire and observational checklist on 43 food handlers (10% of total sample size), then using the pretest result, amendments were done for unclear and vague questions, content validity and internal reliability were checked. If all these are checked and modified before actual data collection, the collected data become qualified; results real study output.

Question #8. if one study participant answer 8 question and missed answer about wearing mask and hand washing how did you accomodate? Did he/she have good knowledge?

Response: Simply we asked Eleven yes/no questions to assess food handlers’ knowledge and we got the minimum score of 2 and maximum score of 11 and the mean score of 9.33 (SD: ±1.97). Base on this mean score, we determined whether he/she had good knowledge or not. Scientifically, 15-20 % of missed value from total questionnaire can be tolerated which means we tolerated if someone give answer for at least 8 out of 11 knowledge questions. As a good news almost all knew about the importance of wearing mask, social distancing and hand washing.

Question #9. You did "Hosmer-Lemeshow test was used to test model fitness. Multicollinearitywas tested using the Variance Inflation Factor (VIF) and tolerance test." Please put thier resul under bracket for Hosmer-Lemeshow test, VIF and tolerance test.

Response: Thank you, and we put the value in bracket. Please see page 11; line number 217-221.

Question #10. Please format randendency one-fifth 96 (23.9%). Put only figure and

percent throughout result section.

Response: Thank you for this remark. We updated the result part as you suggested

Question #11. From 11 questions how did you manage " I do not know" response?

Response: Thank you dear reviewer for your critical thinking but there were no respondents who answered " I do not know"

Question #12. in Attitude section, did you use likert scale(strongly disagreed,agreed

Response: Yes we used nine likert scale attitude statements with the value of each option strongly disagree, disagree, neutral, agree and strongly agree 1, 2, 3, 4 and 5 respectively.

Question #13. Please polish your english and your write up through out your document it needs critical improvements. look your statements e.g. "Most respondents 389 (97%) reported that when they sneezed, they coveredtheir nose and mouth with an elbow. & Of 36 food handlers who did notwear PPE, 10 of them thought that wearing PPE is not always necessary to prevent COVID-19."

Response: Thank you for this comment. We corrected and updated all repetition on the result part.

Question #14. In your discussion part you merged studies and the reason of dicrepancy but you have to discuss in segregated way from global to local and put reason of dicrepancy accordingly.

Response: Thank you for these important comments and we modified the discussion section as you suggested, please see the revised version page 16; line number 311-321.

Question #15. When you reason out the reason of dicrepancy you are not sure about

your conclusion. For example "This may be due to the differences in the tool used for assessment of..." "may be"

Response: Thank you dear reviewer, we changed “This may be due to…” by “The probable reasons might be…”

Reviewer #9: 

Andualem et al., Review: COVID-19 Infection Prevention and Control Practices among Food Handlers of Food and Drinking Establishments of Dessie City and Kombolcha Towns in Northeastern Ethiopia

Dear Editor The manuscript titled “COVID-19 Infection Prevention and Control

Practices among Food Handlers of Food and Drinking Establishments of Dessie City and Kombolcha Towns in Northeastern Ethiopia” presented by Atsedemariam Andualem and co-authors is aimed to highlight the COVID-19 prevention and control practices adopted by food handlers in two cities of Ethiopia. The authors have contributed well by selecting

the important sector of society which can play a significant role towards the transmission and spread of C COVID-19 pandemic. But while reviewing the article there are some ambiguities and questions which needs to be addressed before the article will be assessed for its

publication in PLOS ONE. The details of comments and queries are given below. I hope it will prove to be helpful for the authors. Section wise Comments:

Question #1. Abstract: The first paragraph of Abstract couldn’t successfully generate the hypothesis of the study. The author should clearly write atleast 2 to 3 aims and objectives to justify that why the study has been conducted? The following line is too early to be stated in the beginning therefore should be deleted from here: {Thus, ensuring infection prevention and control practices for this high-risk group is urgently required to prevent COVID-19 transmission}. In the second paragraph of Abstract it is important a brief description should be added about the design of the questionnaire and checklist with its strength and important parameters while the description about statistical tests should be given in one line only. In the last paragraph of Abstract the first line {Just more than one-third of food handlers had good infection prevention and control practices} should be deleted as it seems out of context. The paragraph should be made brief as there is a lot of repetition of words/phrases “infection prevention and control”. It is suggested that only the following text should be given in the last part: {Our findings showed that three-fourths of food handlers had good knowledge and more than half had a favorable attitude about infection prevention and control. Thus, integrated work is urgently needed to prevent COVID-19 transmission by further improving food handlers’ good practice of infection prevention and control}.

Response: Thank you dear reviewer for your important comments, and we modified as you suggested, please see the abstract from revised version of the manuscript line number 20-49.

Question #2. Keywords: should be written in an alphabetical order.

Response: Thank you, we corrected the alphabetical order, please see page 3, line number 50 and 51.

Question #3. Introduction: the first 2- paragraphs should be merged with the following text added at the end of first paragraph while the rest of the text should be deleted as it is only a repetition: {According to a World Health Organization (WHO) report on 20 September 2020, 30.6 million positive cases with 950,000 deaths worldwide have been recorded. On that date, Africa accounted for 1,145,397 COVID-19 positives with a total death of 24,757; and in Ethiopia the number of COVID-19 cases reached 68,131 and confirmed deaths reached 1,089 [9].} In the last paragraph the following sentence should be either deleted or re-written as it does not clearly interpret what is author trying to convey: {Implementation of IPC practices based on WHO and Ethiopian Ministry of Health (EMoH) guidelines is rapid, decisive, and collective action which saves millions of lives [13, 16, 17].}

Response: We updated as your suggestion and please see the introduction part of the revised version from lines 54 to 67and from 87 to 97.

Question #4. The objectives of the study should be elaborated more by focusing on what the authors are trying to achieve by conducting this study and how it will be helpful for the society as the given objective: {Therefore, this study was designed to assess the practice of infection prevention and control strategies against the novel coronavirus pandemic among food handlers in Dessie City and Kombolcha Town food and drinking establishments} is too generalized.

Response: Thank you for this remark. We elaborate the objective of the study, and please see page 5, line number 91-94.

Question #5. Methods and Materials:

Study Area: Add some description about the socioeconomic status of the area to generate some idea about the general people perceptions and attitudes in the study area. Study design, period, and population: The terms “The source population” and “The study population” leads to

confusion as they both refer to study subjects therefore it would be better to use “The study population” to avoid confusion. The following lines should be re-written as seems a repetition {The source population was all food handlers working in Dessie City and Kombolcha

Town food and drinking establishments. The study population was food handlers among the selected food and drinking establishments}. It is also important that detail data in the form of table/graph (could be added in Annexure) should be given about the categories

based on nature of food and drinking facilities included in the study.

Response: Thank you for these comments.

- We included the socioeconomic status of the area, please see page 6; line number 104-110.

- “The source population was all food handlers working in Dessie City and Kombolcha Town food and drinking establishments” is deleted to make it clear.

Question #6. Sample size determination and sampling techniques:

How the sample size from each study area has been selected should be written more clearly as currently it is very ambiguous and unclear. Similar is true for sampling techniques and should be re-written.

Response: Sorry for the confusion we did. Now we make it clear and easily understandable, please see the revised version on page 7, and line number 118-136.

Question #7. Measurement of the outcome variable:

It is important to discuss questionnaire and important aspects included in its structure and how the mean value has been calculated. As currently the most important tool of the study “Questionnaire design” and “checklist” has been lacking from the study which is a serious discrepancy. Operational definitions: These definitions should be given in the Annexure. Data collection and quality control: This part could also be given in the Annexure.

Response: Thank you for these invaluable comments. We correct the comments as you suggested, please see the revised version on page 8 line 146-151 and page 9& 10 line 175-185.

Question #9. Results:

Overall, the results have been interpreted in detail but there is a need to improve the quality of graphs and tables.

Response: Thank you for your suggestion, and we omit the graph to decrease repetition and modified tables.

Question #10. Discussion:

At large Discussion is well written and has thoroughly justified the results. Overall, some grammatical mistakes have been found in the write up along with repetition of words and phrases therefore it is highly recommended that authors should proof read the draft from an expert

with strong English language background to improve the overall quality of the manuscript.

Response: Thank you dear reviewer, all language errors and problems in the whole document are well corrected and addressed.

Reviewer #10: 

The article of Metadel et al. is a completed study and is of interest to scientists for studying the problems of COVID-19 iinfection prevention and control practices among food handlers. The

manuscript describes the the knowledge, practice and attitudes of handlers of food and drinking establishments’ in cities and towns of Ethiopia The paper is well structured but it needs some modification prior to publication. I think this manuscript should be accepted for

publication after some revision: 

Question #1. It is not clearly written in methods, which variables were used for the univariate or multivariate models as interest variables or for the model adjustment.

Response: Sorry for the confusion, and we correct the comments, please see page 8 line 146-151 and page 9& 10 line 175-185.

Question #2. Additionally, for the bivariate analysis, authors could have included variables with p-value up to 0.10 with backward or forward elimination or Other? Why P-value up to 0.1? Do you have reference?

Response: Typical stopping rules for explanatory modeling are p-value thresholds of 0.05 and 0.10. If a p-value is greater than the threshold, the term is removed from the model with backward LR (for our study). 

Our reference is available online at www.jmp.com/en_us/statistics-knowledge-portal/what-is-multiple-regression/variable-selection.

Question #3. Authors must provide more details on methods: which factors and explanatory variables were addressed by the questionnaire and in the analyses (e.g., as continuous or dichotomous).

Response: We correct the comments as you suggested, please see the revised version on page 8 line 146-151 and page 9 & 10 line 175-185.

Question #4. The manuscript has used inappropriate English word “illiterate” which is erroneous that must be removed.

Response: Thank you dear reviewer for the comment, and we replaced illiterate by unable to read and write.

Question #5. It was a kind of cross sectional study, why you used prolonged time (two months) for data collections?

Response: You are right dear reviewer but we used two months for data collection due to COVID-19 pandemic which increases the burden.

Question #6. Why the experiences were defined as <1, >1-5 and >5 years? It should be clearly stated in methods.

Response: Work experiences of food handlers were classified as <1, 1-5 and >5 years based on other previous study’s classification (Desta, 2010).

.

Reviewer #11: There are some observations made in the process of this review.

Question #1. Method: 

The authors mentioned in the section of measures of measures that the mean value from fifteen questions was considered good infection prevention and control practice. The authors should state the score of each question to prevent ambiguity. Readers will be well informed of the for instance, the authors says either each question contains 1,3,3 or so score, from which the mean value I'd calculated. In the data collection and quality control section, authors stated that, protested questions and on the spot observations checklist were used. However, they failed readers of what was entailed on the observations checklist. Readers will appreciate it much, if they are informed what the observations were to be, and anticipate what the observed findings were in the results section

Response:

- We correct the comments as you suggested, please see the revised version on page 8 line 151-156 and page 10 line 180-188.

- From data collection and quality control: we added what the observations to be; please see page 10 line number 181 and 182 and from result section: we put ‘observation’ in bracket to differentiate observation checklist from interview at Table 2.

Question #2. The results: 

The authors did not indicate the observations made by way of the

observations checklist that was employed. instead the results were focused on only the structure question. It is important for readers to know whether answers provide by the study subjects were in concordance with the observations made by the researchers. This will probably

prevent some bias introduced by the subjects because of the researchers' presence. In reporting the the results, the third sentence should be checked and rephrase...." most of the respondents were under 30 years 273 (68.1%)." "The odds of having good COVID-19 infection prevention and control practice among food handlers who had work experience of greater than 5

years were 3 times (AOR=2.55; CI:1.43-5.77) greater compared to those who had 5 or fewer years of work experience. " this statement should be written to reflect the results. It is emphatic, however, that is not the case. Trying saying "almost 3 times....."

Response: Thank you for these pertinent comments.

- From the result section: we put ‘observation’ in bracket to differentiate observation checklist from interview at Table 2.

- In reporting the results, the third sentence is modified, please see page 12; line number 236.

- We modified the sentence as “Food handlers who had work experience of greater than 5 years were almost 3 times (AOR=2.55;CI:1.43-5.77) more likely to have good COVID-19 infection prevention practice than those who had 5 or fewer years”. Please see page 14 and 15; line number 288-290.

Question #3. Discussion: 

In the second paragraph, the researchers identified some difference in results and attributed same to differences in tools "This may be due to the differences in the tool used for assessment of COVID-19 infection prevention and control practice, differences in study participants where our study focused on food handlers, or a methodological difference such as the difference in the accessibility of COVID-19 infection prevention and control guidelines and types of PPE in the food and drinking establishments. "

It is therefore important to mention the tool used in your study and that used in the other studies which could be the plausible reason.

Response: Thank you dear reviewer for the comments, and we modified the discussion section as you suggested, please see the revised version page 16; line number 311-321.

Thank you for your time in reviewing our paper and we hope that the revised version is acceptable for publication in PLoS ONE. 

Sincerely yours, 

Dr Metadel Adane (PhD in Water and Public Health)

---

## [Decision Letter · Decision Letter 1]

17 Sep 2021

PONE-D-21-12395R1COVID-19 Infection Prevention Practices among Food Handlers in Food and Drink Establishments of Dessie City and Kombolcha Town, Northeastern EthiopiaPLOS ONE

Dear Dr. Adane,

Thank you for submitting your manuscript to PLOS ONE. After careful consideration, we feel that it has merit but does not fully meet PLOS ONE’s publication criteria as it currently stands. Therefore, we invite you to submit a revised version of the manuscript that addresses the points raised during the review process.

Great effort was made by the authors to utilize the feedback that was provided for them to correct for resubmission. There are still major things to adjust. Although this topic is an important one, yet the authors did not adopt appropriate methodology to achieve the aim. Accordingly, fundamental issues are indicated to be revised mainly for the methodology section mainly the statistical analysis (enclosed). 

The manuscript could be greatly strengthened by considering editing according to the specific Reviewers’ comments.

We look forward to receiving your revised manuscript.

Kind regards,

Ammal Mokhtar Metwally, Ph.D (MD)

Academic Editor

PLOS ONE

Journal Requirements:

Additional Editor Comments (if provided):

Reviewers' comments:

Reviewer's Responses to Questions

**Comments to the Author**

1. If the authors have adequately addressed your comments raised in a previous round of review and you feel that this manuscript is now acceptable for publication, you may indicate that here to bypass the “Comments to the Author” section, enter your conflict of interest statement in the “Confidential to Editor” section, and submit your "Accept" recommendation.

Reviewer #2: (No Response)

Reviewer #4: (No Response)

Reviewer #6: All comments have been addressed

Reviewer #7: All comments have been addressed

2. Is the manuscript technically sound, and do the data support the conclusions?

Reviewer #2: Partly

Reviewer #4: Partly

Reviewer #6: Yes

Reviewer #7: Yes

3. Has the statistical analysis been performed appropriately and rigorously? 

Reviewer #2: No

Reviewer #4: No

Reviewer #6: Yes

Reviewer #7: I Don't Know

4. Have the authors made all data underlying the findings in their manuscript fully available?

Reviewer #2: Yes

Reviewer #4: No

Reviewer #6: Yes

Reviewer #7: Yes

5. Is the manuscript presented in an intelligible fashion and written in standard English?

Reviewer #2: Yes

Reviewer #4: Yes

Reviewer #6: Yes

Reviewer #7: Yes

6. Review Comments to the Author

Reviewer #2: The authors responded to my suggestion on considering a higher cut-off score to define "good", that the mean is typically the best measure of central tendency because it takes all values into account (the distribution of the data is normal). I believe the authors may have misunderstood what I meant.

If the maximum possible score was 100, and the average mean score was just 20, it is not reasonable to assume that those who scored above 20 are all considered to be "good". However, if the mean score was 90 out of 100, potentially those who scored below 90 but say above 70 would also be considered to have done well. So the mean is not a good way to define "good".

The definition of good should be based on a proportion of the maximum possible score. I think 70 out of 100 is considered "good". Possibly 60 out of 100 could also be "good". I acknowledge that there may be no studies out there which can be referenced to set the threshold definition for good. But surely, 50% cannot be considered "good".

I urge the authors to consider setting the threshold for the definition of "good" at 70% (which would be a score of 18.75) or at 60% (which would be a score of 15). Otherwise, the authors would have to amend their manuscript throughout to state that they "analyzed factors associated with participants who scored above the mean.", which in no way suggests whether these are considered "good" or otherwise.

Reviewer #4: 1- This manuscript has not been prepared from the right perspective of the research methodology.

2- I don’t believe that the sample size calculation is relevant as the underlying primary aim is the multivariate regression modeling

3- Dichotomising the fifteen-infection scale at an arbitrary cut-off (suing the sample mean) reduces the statistical power by the same amount as would discarding a third of the data (1, 2). In general, there is no good reason to believe that there is an underlying dichotomy, and if one exists there is no reason why it should be at the mean especially as the fifteen-infection scale is not normally distributed. The fifteen-infection scale looks like an ideal candidate for one of the ordinal cumulative probability models. Therefore, I suggest using the proportional odds model.

Reviewer #6: Dear authors,

Thank you for allowing me to review this manuscript. I think this is a good study and worth for publication. Generally, I am satisfied with all the corrected version from the manuscript. Keep up the good work. Thanks

Reviewer #7: The article seems clearer with the changes you have made. Thank you for taking my suggestions into consideration.

7. PLOS authors have the option to publish the peer review history of their article (what does this mean?). If published, this will include your full peer review and any attached files.

Reviewer #2: No

Reviewer #4: No

Reviewer #6: No

Reviewer #7: No

---

## [Author Response · Author response to Decision Letter 1]

27 Sep 2021

Date: 25 September 2021

Manuscript ID: PONE-D-21-12395R1

COVID-19 Infection Prevention Practices among Food Handlers in Food and Drinking Establishments of Dessie City and Kombolcha Town in Ethiopia 

Corresponding author: Metadel Adane(PhD)

Dear Dr Ammal Mokhtar Metwally, (PhD)

Academic Editor

PLOS ONE

This paper has been reviewed 11 reviewers for the first time. And again the revised version reviewed by four reviewers. This two rounds of review has made the paper highly improved and we thank you for your letter with a decision of revision needed. We were pleased to know that our manuscript was considered potentially acceptable for publication in PLoS ONE, subject to adequate revision as requested by you. 

We agree with almost all the comments/questions raised by you four reviewers of the second comment. We would like to take this opportunity to express our thanks to you and the reviewers for your valuable comments and to thank you for allowing us to resubmit a revision of the manuscript. 

I hope that the revised manuscript is accepted for publication in PLoS ONE. 

Sincerely yours,

MetadelAdane (PhD)

---

## [Decision Letter · Decision Letter 2]

22 Oct 2021

PONE-D-21-12395R2

COVID-19 Infection Prevention Practices among Food Handlers in Food and Drink Establishments of Dessie City and Kombolcha Town, Northeastern Ethiopia

PLOS ONE

Dear Dr. Adane,

Thank you for submitting your manuscript to PLOS ONE. After careful consideration, we feel that it has merit but does not fully meet PLOS ONE’s publication criteria as it currently stands. Therefore, we invite you to submit a revised version of the manuscript that addresses the points raised during the review process.

We look forward to receiving your revised manuscript.

Kind regards,

Lucinda Shen, MSc

Staff Editor

on behalf of 

Ammal Mokhtar Metwally, Ph.D (MD)

Academic Editor

PLOS ONE<o:p></o:p>

Journal Requirements:

Additional Editor Comments (if provided):

Great effort was made by the authors to utilize the feedback that was provided for them to correct for resubmission after revision. Meanwhile, minor modifications are required according to PLOS one guidelines:

Please consider restricting the Abstract to no more than 300 wordsThe authors declared that oral consent was sought from participants before the study started, please justify why authors did not take written consent and please clarify whether the institutional review board approved the use of verbal consent.Please also state in the ethics approval the approval number 

in addition refer to the journal’s other publication criteria (https://journals.plos.org/plosone/s/criteria-for-publication)

Reviewers' comments:

Reviewer's Responses to Questions

**Comments to the Author**

1. If the authors have adequately addressed your comments raised in a previous round of review and you feel that this manuscript is now acceptable for publication, you may indicate that here to bypass the “Comments to the Author” section, enter your conflict of interest statement in the “Confidential to Editor” section, and submit your "Accept" recommendation.

Reviewer #2: All comments have been addressed

Reviewer #6: All comments have been addressed

2. Is the manuscript technically sound, and do the data support the conclusions?

Reviewer #2: Yes

Reviewer #6: Yes

3. Has the statistical analysis been performed appropriately and rigorously? 

Reviewer #2: Yes

Reviewer #6: Yes

4. Have the authors made all data underlying the findings in their manuscript fully available?

Reviewer #2: Yes

Reviewer #6: Yes

5. Is the manuscript presented in an intelligible fashion and written in standard English?

Reviewer #2: Yes

Reviewer #6: Yes

6. Review Comments to the Author

Reviewer #2: (No Response)

Reviewer #6: Thank you for allowing me to review this manuscript. I think this is a good study and worth for publication. Generally, I am satisfied with all the corrected version from the manuscript. Everything already improvised.

Keep up the good work.

7. PLOS authors have the option to publish the peer review history of their article (what does this mean?). If published, this will include your full peer review and any attached files.

Reviewer #2: No

Reviewer #6: No

---

## [Author Response · Author response to Decision Letter 2]

23 Oct 2021

Rebuttal letter

Response to Editor questions/comments for the paper PONE-D-21-12395R2

Dear Editor, 

Thank you for your pertinent comments and we updated the manuscript accordingly.

Here are the detail answers for your concerns: 

Additional Editor Comments (if provided):

Great effort was made by the authors to utilize the feedback that was provided for them to correct for resubmission after revision. Meanwhile, minor modifications are required according to PLOS one guidelines:

Response: Thank you for your positive remark and for your decision of minor revision required. We updated the manuscript accordingly and please find the detail response for each hereunder. 

1. Please consider restricting the Abstract to no more than 300 words

Response: Thank you. We updated the manuscript to 300 words. 

2. The authors declared that oral consent was sought from participants before the study started, please justify why authors did not take written consent and please clarify whether the institutional review board approved the use of verbal consent.

Response: Sorry for this confusion. We used written informed consent, not oral consent. We updated the manuscript. 

3. Please also state in the ethics approval the approval number

Response: Thank you for this important comment. We included the approval number in the ethics statement.

---

## [Editor Report · Decision Letter 3]

28 Oct 2021

COVID-19 Infection Prevention Practices among Food Handlers in Food and Drink Establishments of Dessie City and Kombolcha Town, Northeastern Ethiopia

PONE-D-21-12395R3

Dear Dr. Adane,

We’re pleased to inform you that your manuscript has been judged scientifically suitable for publication and will be formally accepted for publication once it meets all outstanding technical requirements.

Kind regards,

Ammal Mokhtar Metwally, Ph.D (MD)

Academic Editor

PLOS ONE

Additional Editor Comments (optional):

The authors have done a good work by addressing all the comments raised in the previous review.
---

## [Editor Report · Acceptance letter]

13 Jan 2022

PONE-D-21-12395R3 

COVID-19 Infection Prevention Practices among a sample of Food Handlers of Food and Drink Establishments in Ethiopia 

Dear Dr. Adane:

I'm pleased to inform you that your manuscript has been deemed suitable for publication in PLOS ONE. Congratulations! Your manuscript is now with our production department. 

Kind regards, 

on behalf of

Professor Ammal Mokhtar Metwally 

Academic Editor

PLOS ONE